# Blood Proteome Profiling Reveals Biomarkers and Pathway Alterations in Fragile X PM at Risk for Developing FXTAS

**DOI:** 10.3390/ijms241713477

**Published:** 2023-08-30

**Authors:** Marwa Zafarullah, Jie Li, Michelle R. Salemi, Brett S. Phinney, Blythe P. Durbin-Johnson, Randi Hagerman, David Hessl, Susan M. Rivera, Flora Tassone

**Affiliations:** 1Department of Biochemistry and Molecular Medicine, School of Medicine, University of California Davis, Sacramento, CA 95817, USA; mzafarullah@ucdavis.edu; 2Genome Center, Bioinformatics Core, University of California Davis, Davis, CA 95616, USA; jjsli@ucdavis.edu; 3Genome Center, Proteomics Core, Genome and Biomedical Sciences Facility, University of California Davis, Davis, CA 95616, USA; msalemi@ucdavis.edu (M.R.S.); bsphinney@ucdavis.edu (B.S.P.); 4Division of Biostatistics, School of Medicine, University of California Davis, Davis, CA 95616, USA; bpdurbin@ucdavis.edu; 5MIND Institute, University of California Davis Medical Center, Sacramento, CA 95817, USA; rjhagerman@ucdavis.edu (R.H.); drhessl@ucdavis.edu (D.H.); smrivera@umd.edu (S.M.R.); 6Department of Pediatrics, University of California Davis Medical Center, Sacramento, CA 95817, USA; 7Department of Psychiatry and Behavioral Sciences, University of California Davis Medical Center, Sacramento, CA 95817, USA; 8Department of Psychology, University of California Davis, Davis, CA 95616, USA; 9Department of Psychology, University of Maryland, College Park, MD 20742, USA

**Keywords:** fragile X-associated tremor/ataxia syndrome, FXTAS, premutation, blood proteomic, biomarker, protein alterations, pathways

## Abstract

Fragile X-associated Tremor/Ataxia Syndrome (FXTAS) is a neurodegenerative disorder associated with the *FMR1* premutation. Currently, it is not possible to determine when and if individual premutation carriers will develop FXTAS. Thus, with the aim to identify biomarkers for early diagnosis, development, and progression of FXTAS, along with associated dysregulated pathways, we performed blood proteomic profiling of premutation carriers (PM) who, as part of an ongoing longitudinal study, emerged into two distinct groups: those who developed symptoms of FXTAS (converters, CON) over time (at subsequent visits) and those who did not (non-converters, NCON). We compared these groups to age-matched healthy controls (HC). We assessed CGG repeat allele size by Southern blot and PCR analysis. The proteomic profile was obtained by liquid chromatography mass spectrometry (LC-MS/MS). We identified several significantly differentiated proteins between HC and the PM groups at Visit 1 (V1), Visit 2 (V2), and between the visits. We further reported the dysregulated protein pathways, including sphingolipid and amino acid metabolism. Our findings are in agreement with previous studies showing that pathways involved in mitochondrial bioenergetics, as observed in other neurodegenerative disorders, are significantly altered and appear to contribute to the development of FXTAS. Lastly, we compared the blood proteome of the PM who developed FXTAS over time with the CSF proteome of the FXTAS patients recently reported and found eight significantly differentially expressed proteins in common. To our knowledge, this is the first report of longitudinal proteomic profiling and the identification of unique biomarkers and dysregulated protein pathways in FXTAS.

## 1. Introduction

The prevalence of various neurodegenerative diseases, such as Alzheimer’s dementia and Parkinson’s disease, has risen in recent years among many populations due to the increase in the aging population. Developing effective treatments for these complex disorders is challenging due to the complex underlying molecular mechanisms involved, the lack of biomarkers for early diagnosis, the broad spectrum of symptoms, limited natural history, data, and the difficulty in conducting clinical trials with small patient populations. Identifying biomarkers and changes in the associated pathways, particularly in assays, that can quickly and objectively indicate changes in disease pathology is crucial for improving patient outcomes.

Fragile X-associated Tremor/Ataxia Syndrome (FXTAS) is a late-onset neurodegenerative disorder with an average age of onset of 62 that affects carriers of a premutation (PM) allele (55–200 CGG repeats) in the fragile X messenger ribonucleoprotein 1 (*FMR1*) gene, usually presenting with a more severe clinical phenotype in males, likely due to the presence of a second X chromosome in females [1,2]. The high prevalence of the premutation allele among the general population (1:430 males and 1:110–200 females) leads to an estimate of approximately 1.5 million individuals in the general US population who are at risk for *FMR1* premutation associated disorders over their life spans. In addition, among the PM population, an estimated 8–16% of females and 40–60% of males are at risk of developing FXTAS [2,3].

Currently, there is no effective specific treatment for FXTAS, and the motor/cognitive symptoms progressively worsen over time, causing reduced quality of life, increased medical expenses, and eventually premature death. FXTAS is clinically distinguished by the presence of intention tremor, cerebellar ataxia, global brain atrophy and white matter disease, autonomic dysfunction, progressive Parkinsonism, and ubiquitin-positive intranuclear inclusions in brain astrocytes, neurons, and Purkinje cells [4]. It is caused by the expanded CGG repeats (55–200 CGG) in the 5′UTR of the *FMR1* gene. In those with the normal *FMR1* gene, the number of CGG repeats lies between 5 and 44, while individuals carrying alleles with a repeat expansion greater than 200 develop fragile X syndrome (FXS), the most common form of intellectual disability and monogenic cause of autism spectrum disorder (ASD) [5]. At the molecular level, the eight- to tenfold increase in the level of *FMR1* mRNA in a PM containing the expanded CGG repeats [6] leads to RNA toxicity and ultimately to neurodegeneration. Three main mechanisms have been proposed to explain the pathogenesis of FXTAS, including the sequestration of CGG-binding proteins amplified by the elevated levels of *FMR1* mRNA, the production of toxic FMRPolyG proteins due to RAN translation, and the chronic activation of the DNA damage response [7,8].

Mass spectrometry (MS)-based proteomics, which involves the advance of data mining and bioinformatic analysis to examine protein structure and function, can be used as an effective technology to quickly analyze large amounts of clinical and biological information within a given sample [9]. Recent advances in proteomic profiling technology and processing have also made it possible to efficiently analyze hundreds of proteins, precisely obtain a snapshot of the altered pathways in an organism and identify biomarkers for disease development and progression [10]. Although these MS-based proteomic workflows for biomarker discovery and profiling are well established, studies focused on blood proteome profiling and, importantly, on samples collected at different time points have not been carried out in PM at risk of FXTAS.

Recently, Ma and colleagues (2019) performed LC-MS/MS-based proteomics of the intranuclear inclusion isolated from postmortem FXTAS brain tissue. Their work highlighted the presence of more than 200 proteins within the inclusions, including a high abundance of SUMO2 and p62/sequestosome-1 (p62/SQSTM1), supporting a model where the inclusion formation results from increased protein loads and elevated oxidative stress [11]. Later, based on these observations, a proteomic profile was characterized in the FXTAS cortex as compared to those obtained from healthy controls (HC) [12]. Specifically, a significant decrease in the abundance of proteins including tenascin-C (TNC), cluster of differentiation 38 (CD38), and phosphoserine aminotransferase 1 (PSAT1) was observed in these samples. In addition, the authors confirmed the significantly high abundance of novel neurodegeneration-related proteins and of the small ubiquitin-like modifier 1/2 (SUMO1/2) in the FXTAS cortex as compared to HC [12]. Finally, a recent study reported changes in the levels of many proteins, including amyloid-like protein 2, contactin-1, afamin, cell adhesion molecule 4, NPC intracellular cholesterol transporter 2, and cathepsin, by comparing the cerebrospinal fluid (CSF) proteome of FXTAS patients with the CSF of HC patients. Changes in acute phase response signaling, liver X receptor/retinoid X receptor (LXR/RXR) activation, and farnesoid X receptor (FXR)/RXR activation pathways were observed [13].

Importantly, no study evaluating predictive biomarkers by blood proteomic alterations in PM, who developed symptoms of FXTAS over time has been reported to date. Here, we present our findings on global profiling derived from male participants enrolled in an ongoing longitudinal study carried out at the UC Davis MIND Institute. The participants have been followed for at least two longitudinal time points: Visit 1 (V1) and Visit 2 (V2). At each time point, neuroimaging, neuropsychological, and molecular measurements, as well as medical and neurological examinations, were collected. A fraction of the premutation participants, all symptom-free or not meeting criteria for FXTAS diagnosis at the time of enrollment (V1), developed symptoms later on (V2) that warranted a diagnosis of FXTAS during the study and were defined as converters (CON). The remaining premutation participants who did not develop symptoms that warranted a diagnosis of FXTAS by the time of the follow-up visit at (V2) are here defined as non-converters (NCON). In the current work, we performed the blood proteome profiling of PM, including CON and NCON, at both V1 and V2 and compared it to HC. We identified a number of potential predictive proteomic biomarkers for early diagnosis, as they showed significant changes in expression levels over time only in the converter group, and we also reported the altered protein pathways among the groups, suggesting their role in the pathogenies of the disorder.

## 2. Results

### 2.1. Demographics

DNA testing confirmed the presence of a premutation allele in the PM group, with the participants who converted at V2 (CON; *n* = 17) and PM who did not convert at V2 (NCON; n = 19), and the absence of a premutation allele in the healthy control (HC; n = 12) group. Participant ethnicity did not differ significantly between the three groups. CGG repeat numbers were significantly lower in healthy controls compared to the other two groups (*p* < 0.001 in both comparisons) but not significantly different between CON and NCON. Healthy controls were significantly younger than non-converters (*p* = 0.0319), as reported in Table 1.

### 2.2. Differential Protein Expression between Healthy Control and Premutation Groups

To identify biomarkers potentially associated with the development and progression of FXTAS, we compared the blood proteomic profile of HC to the PM, including CON and NCON. The groups display a separation trend, as shown in Figure 1. A sparse partial least squares discriminant analysis (s-PLSDA) was performed, which showed that all samples from each group aggregated, and the separation between groups indicated differences in the proteomic characteristics between PM and HC and between CON and NCON. A total of 79 proteins were identified by s-PLSDA analysis to be features that separate the groups. Out of these, 78 were among the list of significantly differentially expressed proteins in differential expression analysis using limma. Their expression profile is summarized in Table 2 and Figure 2.

### 2.3. Identification of Proteomic Biomarkers of FXTAS

From this untargeted proteomic profiling, we identified 227 proteins that showed significant changes in expression (adjusted *p* < 0.05) in pairwise comparisons of the CON as compared to the NCON at V1 (Table 3) and 196 proteins at V2 (Table 4). Between the CON and NCON, we observed 67 proteins that were consistently differentially expressed (adjusted *p* < 0.05) at V1 and kept changing at V2 (Table 5). While comparing the visits, we identified 170 differentially expressed (adjusted *p* < 0.05) proteins between V1 and V2 in the converter group, suggesting their role as biomarkers for the progression of FXTAS (Table 6).

### 2.4. Protein, Lipids, and Amino-Acid Pathways Altered in Individuals Who Developed FXTAS over Time

We further identified the pathways that are altered from V1 to V2 in CON and NCON, including protein lipids and amino acids. Upon examination of protein pathways that were altered between visits in NCON and CON (Figure 3), we found that pathways associated with cell signaling, immune function, cellular organization growth and proliferation, and inflammatory response were those that were more significantly altered from V1 to V2 in the CON group. Similarly, when investigating the protein pathways altered between NCON and CON at V1 or V2, we found that pathways related to synapse signaling (retrograde endocannabinoid signaling pathway) and lipid metabolism were more significantly altered between NCON and CON at V2 (Figure 4). Interestingly, when investigating the list of consistently differentially expressed proteins between CON and NCON groups at V1 and V2, we observed that the pathways related to neurodegeneration are ranked among the top enriched pathways, including the pathways of neurodegeneration, Huntington’s disease, and Alzheimer’s disease (Figure 5), which provides confidence that the potentially relevant biomarkers may be among these proteins. From the gene ontology point of view, the proteins that are consistently differentially expressed between CON and NCON at both visits are enriched in mitochondrial functions, protein synthesis machinery, and transport, as well as positive regulation of the BMP signaling pathway (Figure 6). These suggest the association of this list of proteins with FXTAS development, similar to other neurodegenerative disorders. Further, upon development of FXTAS at V2 (Figure 7), we observed a high level of dysregulation in retrograde endocannabinoid signaling pathways, mRNA surveillance pathways, cancer, cGMP-PKG signaling, calcium, sphingolipid, and lipid pathways, as observed in other neurodegenerative disorders such as Alzheimer’s disease, dementia, and Parkinsonism [14]. Further investigating the lipid and amino-acid metabolism, we detected various associated proteins that were differentially expressed in CON as compared to NCON at V2, suggesting their role in the progression of FXTAS (Figure 8).

### 2.5. Differentially Expressed Common Proteins Identified from CSF and Blood Proteomic Profiling

We compared the blood proteome profile of CON at V2 with the recently reported cerebrospinal fluid (CSF) proteome of FXTAS patients. The CSF proteome identified 414 proteins, out of which 46 were identified to be significantly altered between FXTAS patients and controls [13]. In the present study of the blood proteome, we identified a total of 2166 proteins, of which 97 were found to be common with the CSF proteome, and eight proteins were significantly altered in both studies, including Complement C3, Alpha-2-HS-glycoprotein, Pigment epithelium-derived factor, Inter-alpha-trypsin inhibitor heavy chain H2, Retinol-binding protein 4, Alpha-2-macroglobulin, Prothrombin, and Lumican (Figure 9).

## 3. Discussion

The identification of protein biomarkers and altered molecular pathways in FXTAS is a crucial requirement for both the research and clinical communities as it improves our ability to identify individuals most at risk for the disease as well as to create novel targeted therapies. There are multiple proteins and pathways that were found to be highly implicated in FXTAS. SUMO2 and p62/sequestosome-1 (p62/SQSTM1) proteins have been observed to accumulate in intranuclear inclusions isolated from postmortem FXTAS brain tissue [11], while tenascin-C (TNC), cluster of differentiation 38 (CD38), and phosphoserine aminotransferase 1 (PSAT1) have been observed in FXTAS cortex [12]. Furthermore, it is worth noting that previous studies have examined the proteomic profile of cerebrospinal fluid (CSF) in individuals with Fragile X-associated Tremor/Ataxia Syndrome (FXTAS), highlighting alterations in proteins and pathways when compared to healthy controls [13]. However, to the best of our knowledge, our study represents the first longitudinal investigation of blood proteomic changes specifically in PM, some of whom exhibit progressive symptoms of FXTAS over time. These findings provide valuable insights into the potential role of these proteomic alterations as biomarkers for early diagnosis, disease progression, and the overall development of FXTAS.

We observed a number of important proteins altered between HC and PM, including both CON and NCON (Table 2). Further, we found that a number of those proteins associated with various important pathways are dysregulated between CON and NCON at V1 (Table 3), V2 (Table 4), and even between visits (Table 5 and Table 6). Interestingly, most of these significantly dysregulated proteins are linked to essential pathways and reported to be involved in the development of other age-related neurodegenerative disorders like Alzheimer’s disease, dementia, and Parkinsonism.

In our previous study, we reported lipid and amino acid metabolism dysregulation along with mitochondrial dysfunction in individuals developing FXTAS over time. Specifically, we reported on the clear involvement of different types of lipids in FXTAS and provided evidence of the role that their dysregulation plays in the development and progression of FXTAS [15,16]. Specifically, we have identified altered sphingolipid metabolic pathways, including increased levels of sphingosine, sphinganine, and ceramides, in PM who developed FXTAS over time. Further, we reported on decreased levels of the hexosylceramides and lactosylceramides (LCER), both implicated in neuroinflammatory diseases and mitochondrial dysfunction [17,18], common features observed in FXTAS. In this study, we confirmed and validated the previous finding as we observed abrupted sphingolipids and amino acid metabolism (Figure 8) along with mitochondrial dysfunction in PM, including both CON and NCON at the protein level (Figure 6).

Indeed, proteomic profiles clearly show a different protein signature among the groups (CON vs. NCON at both V1 and V2), and enrichment pathway analysis demonstrates the involvement of key pathways, including lipids, mitochondria, neurodegeneration, and others, as illustrated in Figure 5, Figure 6, Figure 7 and Figure 8. Among these proteins, the cytochrome c oxidase subunit Va (COX5A) and the mitochondrial electron transport chain associated protein MT-CO2 were differentially expressed in the CON group (Table 6). As COX5A is involved in maintaining normal mitochondrial function and plays a vital role in aging-related cognitive deterioration via BDNF/ERK1/2 regulation [19], it could represent a potential target for anti-senescence drugs. The mitochondrially encoded cytochrome C oxidase II (MT-CO2) is located in the mitochondrial inner membrane is part of the respiratory chain complex IV, which is defective in individuals with FXTAS. It is a biomarker of Huntington’s disease [20] and associated with cerebellar ataxia and neuropathy [21], both clinical features observed in FXTAS. Further, a recent metabolomic study of patients with mitochondrial disease demonstrated elevated acylcarnitine levels, suggesting that an altered fatty acid oxidation pathway may represent a downstream mitochondrial respiratory chain dysfunction [22]. Interestingly, we reported high levels of plasma acylcarnitines in the CON group but not in the NCON group [15].

Neural degeneration is a key contributor to the development of neurodegenerative disorders, and we observed a differential expression of the VASP protein in CON. Downregulation of VASP leads to neuronal cell death through an apoptotic pathway and is implicated in the establishment and maintenance of the axonal structure; changes in the expression level can trigger neuronal degeneration [23]. In addition, we identified the RNA and mRNA protein pathway dysregulation in CON at V2 (Figure 7), including snRNA-associated Sm-like protein (LSm3), a critical activating factor for mRNA removal in eukaryotic cells participating in RNA metabolism, silencing, and degradation. Abnormal expression of LSM3 has been found to be associated with mild cognitive impairment (MCI) and Alzheimer’s disease (AD) [24].

In one of the recent studies, Abbasi and colleagues characterized the cerebrospinal fluid (CSF) proteome of FXTAS patients and reported 317 proteins, among which the expression levels of 38 were significantly altered between FXTAS patients and controls [13]. We looked at the overlap with our dataset of 2069 identified proteins and found 97 proteins in common, along with eight significantly altered proteins in both studies (Figure 9). Retinol-binding protein 4 (RBP4) is one of the 8 proteins altered in the CSF as well as in the blood of individuals who developed FXTAS over time. RBP4 is the sole specific transport protein for vitamin A (retinol), and it has been reported that RBP4 can directly induce retinal neurodegeneration in mice through microglia [25]. In the CON group, we also observed increased levels of the C3 protein, a key component of the complement cascade signaling pathway and of the immune system that plays a crucial role in inflammation and host defense [26]. Overactivation of C3 has also been reported in AD, leading to neuronal damage [27], suggesting its contribution to neurodegeneration in various neurological diseases, including AD and FXTAS. Our findings demonstrate that overactivation of C3 could be contributing to neurodegeneration and that perhaps blocking C3 function could be protective and might lead to the development of strategies for future target treatments.

Among the other proteins, which were commonly differentially expressed in this study using blood from FXTAS (CON V2) and in the study using CSF, were the pigment epithelium-derived factor (PEDF), a unique neurotrophic protein that decreases with aging, the acute-phase protein alpha-2 macroglobulin (A2M), which is a significant component of the innate immune system; the serine protease inhibitor (SERPIN), associated with diverse thrombosis disorders, the inter-alpha-trypsin inhibitor heavy chain 2 (ITIH2), the small leucine-rich proteoglycan (LUM), a member of the small leucine-rich proteoglycan family playing a role in cancer, adhesion, and migration [28,29,30]. These neurodegeneration-associated proteins have also been linked to inflammatory processes [28,31], which are observed in FXTAS pathogenesis and may be promising target pathways for pharmacology.

Finally, blood-brain barrier (BBB) abnormalities have been reported across multiple neurodegenerative disorders such as vascular dementia, MS, Lewy body disease, and spinal muscular atrophy and may contribute to the neurological pathology that often enhances neurodegenerative disorders [32,33]. The CSF/serum quotient of albumin (QAlb) is an indirect measurement of the permeability of the BBB, and [13] highlighted the significant correlations between patients’ QAlb and their respected CGG repeat length and FXTAS rating scale score. They suggested that the observed higher QAlb levels in their study and also in the CON group from our study presented here were associated with a more severe clinical phenotype and proposed dysregulations in BBB permeability as a clinical prognostic measure for disease severity for patients diagnosed with FXTAS. Of relevance, our findings that disruption in protein levels and associated pathways, detected in both blood and CSF, argue in favor of the use of a less invasive tissue, blood, to be utilized to identify molecular biomarkers, predictors of disease development, severity, and progression.

One of the limitations of this study is the small sample sizes; however, it is important to acknowledge that FXTAS is a disease that has been understudied and is not common, making it challenging to obtain a larger sample pool. Despite these obstacles, longitudinal and additional studies with a larger sample size should be conducted to confirm our findings, identify the most robust and predictive biomarkers, and gain further insight into the disease pathogenesis. Despite these limitations, our study offers valuable information on the proteomic differences between PM, who developed the disorder over time and controls, which can lead to more comprehensive research into the disease’s underlying mechanisms and potential therapeutic interventions.

## 4. Materials and Methods

### 4.1. Study Participants

As part of a continuing longitudinal study, male participants PM, over the age of 45 years and male participants with non-carrier age-matched healthy controls (HC) were recruited as detailed in [34]. All participants were white in ethnicity, with the exception of three Hispanic participants in the HC group, one in the CON group, and none in the NCON group. The studies and all protocols were carried out in accordance with the Institutional Review Board at the University of California, Davis. All participants gave written informed consent before participating in the study, in line with the Declaration of Helsinki. FXTAS stage scoring was based on the clinical descriptions as previously described [35]. Three categories were used in the diagnosis of FXTAS as explained in Zafarullah and Tassone [36] and termed “definite”, “probable” and “possible FXTAS”. Three age-matched groups were included in this study: CON, NCON, and HC. Using the data from two brain scans, neurological assessment, FXTAS stage, and CGG repeat length, 17 participants were classified as “CON” as they developed clear FXTAS symptomology and thus met criteria of diagnosis between visits (FXTAS stage score was 0–1 at V1 and ≥2 at V2); 19 were defined as “NCON” because they continued to show no signs of FXTAS at V2 (FXTAS stage score was 0–1 at both V1 and V2); and 12 as HC (normal *FMR1* alleles/non-PM). The stages of FXTAS range from no tremor at stage 0 to significant tremor that interferes with activities of daily living and intermittent falls at stage 3 [35].

### 4.2. CGG Repeat Length

Genomic DNA (gDNA) was isolated from 5 mL of peripheral blood leukocytes using the Gentra Puregene Blood Kit (Qiagen, Hilden, Germany). CGG repeat allele size and methylation status were assessed using a combination of Southern blot and PCR analysis. Details of the protocols are as previously reported [37,38].

### 4.3. Sample Handling and Preparation

Peripheral blood was collected in cell preparation tube (CPT) vacutainers with sodium citrate (Becton Dickinson, Singapore) and centrifuged according to the manufacturer’s recommendations for separating mononuclear cells from whole blood. PBMCs were washed with Dulbecco’s phosphate-buffered saline (PBS) and frozen in RPMI 1640 media with 10% fetal bovine serum and 10% dimethyl sulfoxide. Frozen, isolated PBMCs were quickly thawed in a 37 °C water bath, transferred to a 1.5 mL tube, and spun for 20 min to pellet the cells. The freezing medium was removed, and proteins were extracted in 5% SDS in 50 mM triethyl ammonium bicarbonate (TEAB). Protein concentration was determined by BCA assay (Pierce, Appleton, WI, USA), and 150 ug of proteins was digested on an S-Trap™ (ProtiFi, New York, NY, USA) Digestion column plate. Initially, 10 mM dithiothreitol (DTT) was added, incubated at 50 °C for 10 min, and rested at room temperature for 10 min. Next, 5 mM iodoacetamide (IAA) was added and incubated at room temperature for 30 min in the dark. The samples were acidified with 12% phosphoric acid, followed by the addition of freshly made S-trap buffer (90% methanol, 100 mM TEAB, pH 7.1), and mixed immediately by inversion.

The entire acidified lysate buffer mix was transferred to the S-trap plate and pushed through with a Tecan Resolvex A200 (Tecan, Männedorf, Switzerland) until all the solution passed through. Columns were washed with 400 μL of S-trap buffer. Trypsin enzyme digest buffer was carefully added (1:25 enzyme: total protein in 120 μL of 50 mM TEAB, pH 8.0) to the column. After two hours of incubation at 37 °C, the same amount of trypsin and TEAB was added to the S-trap as a boost step, and the reaction continued overnight at 37 °C. The following day, peptides were eluted from the S-trap. Peptide elution steps included 80 μL of 50 mM TEAB (pH 8.0) and 80 μL of 0.5% formic acid 80 μL of the solution containing 50% acetonitrile and 0.5% formic acid. The final pooled elution was dried down in a speed vacuum. Peptides were resuspended in 0.1% TFA and 2% ACN and quantified using the Pierce™ Quantitative Fluorometric Peptide Assay (Thermo Fisher Scientific, Waltham, MA, USA).

### 4.4. Liquid Chromatography Mass Spectrometry (LC-MS/MS)

LC separation was carried out on a Dionex Nano Ultimate 3000 (Thermo Scientific) with a Thermo Easy-Spray source fitted with a PepSep emitter. The digested peptides were reconstituted in 2% acetonitrile/0.1% trifluoroacetic acid, and 5 µL of each sample was loaded onto a Thermo Scientific PepMap 100 C18 5 μm 0.3 mm × 5 mm reverse phase trap, where they were desalted online before being separated on a PepSep 8 cm ID 150 1.5 μm reverse phase column. Peptides were eluted using a 90 min gradient with a flow rate of 0.500 μL/min. The samples were run on an Orbitrap Exploris 480 (Thermo Scientific) in data-independent acquisition (DIA) mode; mass spectra were acquired using a collision energy of 30, resolution of 30 K, maximum inject time mode on auto, and an AGC target of 1000%, using an isolation window of 45.7 Da in the *m*/*z* range 350–1200 *m*/*z*. Raw spectrometry data and analysis are available from the Massive and Proteome Exchange repositories using the respective ID numbers (MSV000092680, PXD044608).

### 4.5. Data Analysis

DIA data were analyzed using Spectronaut 15 (Biognosys Schlieren, Schlieren, Switzerland), using the direct DIA workflow with the default settings. Briefly, trypsin/P-Specific was set for the enzyme, allowing two missed cleavages. Fixed modifications were set for carbamidomethyl, and variable modifications were set to acetyl (protein N-term) and oxidation. For DIA search identification, PSM and Protein Group FDR were set at 0.01%. A minimum of 2 peptides per protein group were required for quantification. A report was exported from Spectronaut using the reporting feature and imported into SimpliFi (https://simplifi.protifi.com/) for QC and statistical analysis (Protifi, Farmingdale, NY, USA).

For the age and CGG repeats, the *p*-values are from an ANOVA F-test followed by Tukey HSD pairwise comparisons. Differential expression analyses were conducted using limma-voom. For comparisons between PM and HC participants at baseline, the model used in limma included PM/HC as the only factor. For analyses of PM among participants at V1 and V2, the model used in limma included factors for conversion status, time, and the interaction between conversion status and time, and estimates and standard errors of log fold changes were adjusted for within-participant correlations. Multiple testing corrections were carried out using the Benjamini-Hochberg (BH) approach. Pathway enrichment analysis was carried out using the Wilcoxon rank-sum test on the raw *p*-values from the differential expression analysis on individual comparisons. Pathway enrichment analysis was carried out using Fisher’s exact test on the overlapping list of proteins that are significantly different between CON and NCON at V1 and at V2 using the BH adjusted *p*-value cutoff of 0.05. Pathway enrichment visualization uses the continuous raw *p*-values from the enrichment analysis. sPLS-DA analysis was carried out using the R package mixOmics version 3.17 [39].

## 5. Conclusions

Currently, there is no effective treatment for FXTAS, and the only options available focus on managing the symptoms. So, a deep understanding of the FXTAS pathogenesis requires the identification of proteins that can be used to understand the altered pathways, serve as biomarkers for early identification of the most at-risk carriers to develop the syndrome, and lead towards the development of targeted therapeutics. However, the investigation of neurodegenerative disorders, including FXTAS, is limited by the availability of accessible sample types. In this study, by using a unique approach of high-throughput mass spectrometry proteomic profiling of blood samples from PM, including longitudinal analysis, we identified a unique set of potential proteomic biomarkers for early diagnosis of FXTAS. In addition, we also observed a significant dysregulation in various protein pathways involved in cellular function and inflammatory responses. These identified pathways may be valuable for the development of effective drugs and therapeutics for this devasting neurodegenerative disorder.

## Figures and Tables

**Figure 1 ijms-24-13477-f001:**
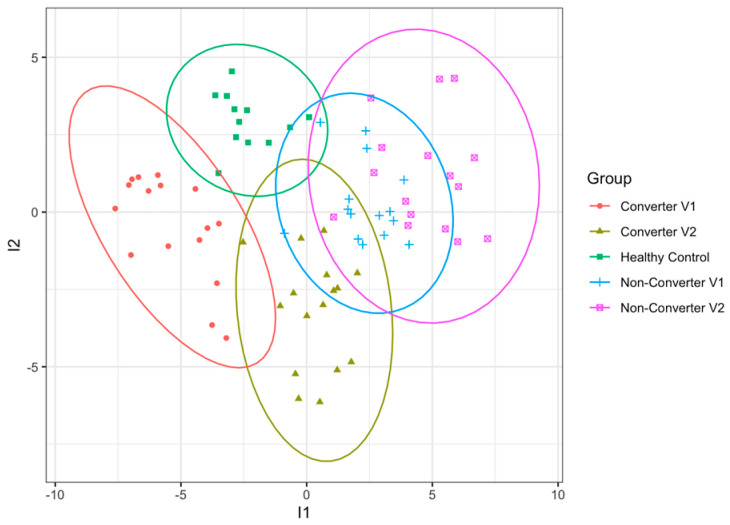
Blood proteome analysis of the present study. The sparse partial least squares discriminant analysis (sparse PLSDA) score plot based on the data of the blood proteome from converters and non-converters (V1 and V2) and healthy controls.

**Figure 2 ijms-24-13477-f002:**
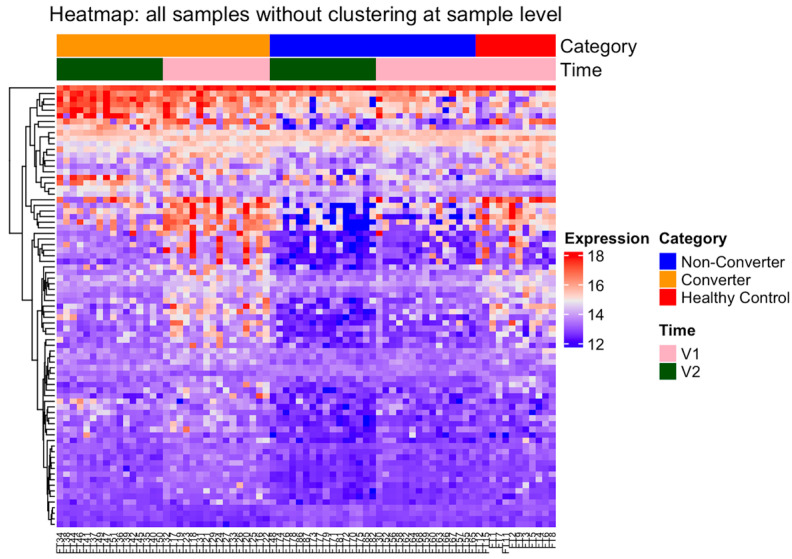
Differential protein expression levels among the HC and the PM groups. Heatmap of the 79 most significantly altered proteins (adjusted *p* < 0.05) in the PM group as compared to HC at both V1 and V2. Heatmap was generated using R code; red indicates high and blue indicates low gene expression.

**Figure 3 ijms-24-13477-f003:**
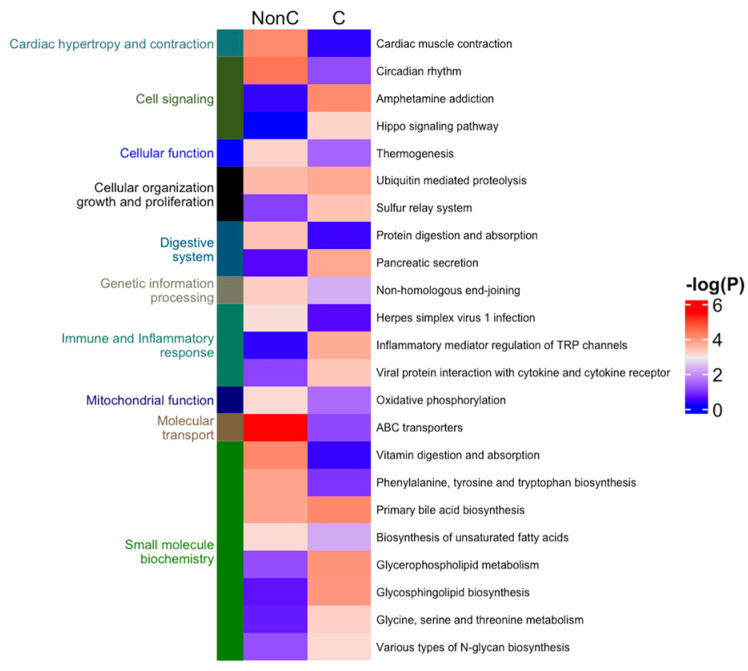
Protein pathways altered from V1 to V2 in CON and NCON groups. Heatmap of the protein pathways that are altered between Visit 2 and Visit 1 in NCON and CON groups. Heatmap was generated using R code; the color from blue to red indicates the increase in statistical significance.

**Figure 4 ijms-24-13477-f004:**
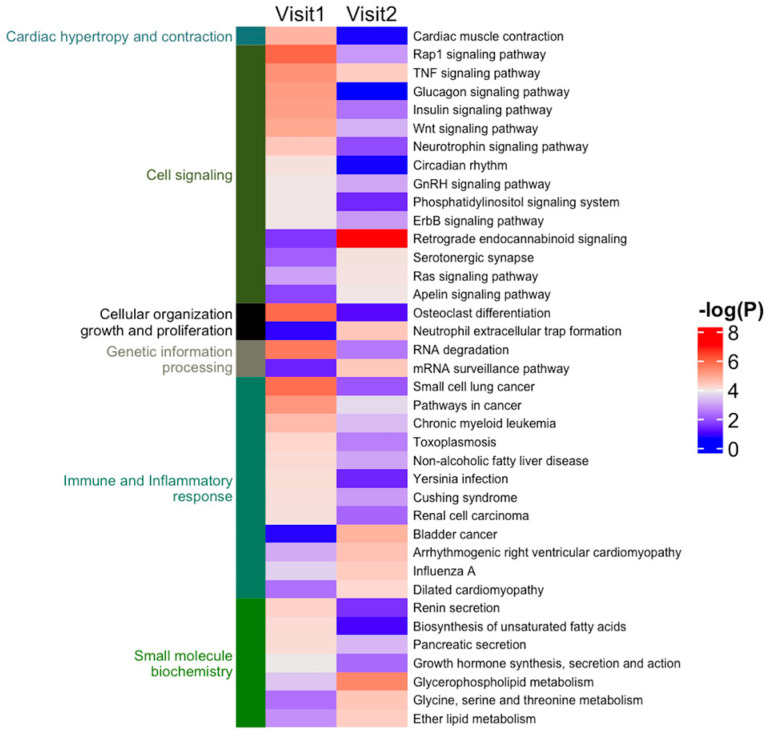
Protein pathways altered between CON and NCON groups. Heatmap of the protein pathways that are altered (*p* < 0.05) between CON and NCON at V1 and V2. Heatmap was generated using R code; the color from blue to red indicates the increase in statistical significance.

**Figure 5 ijms-24-13477-f005:**
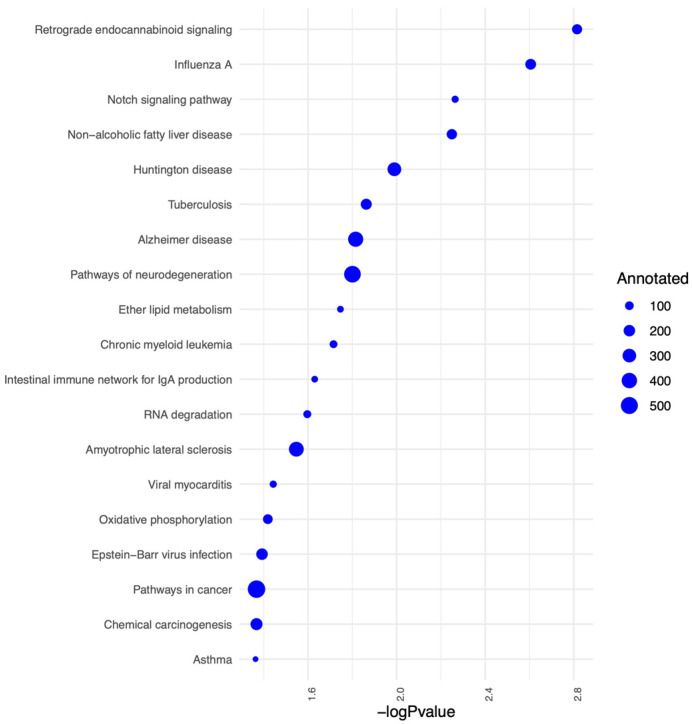
Enriched pathways for the proteins that are consistently differentially expressed between CON and NCON from V1 to V2. Protein-protein interactions from STRING database are represented as edges between proteins.

**Figure 6 ijms-24-13477-f006:**
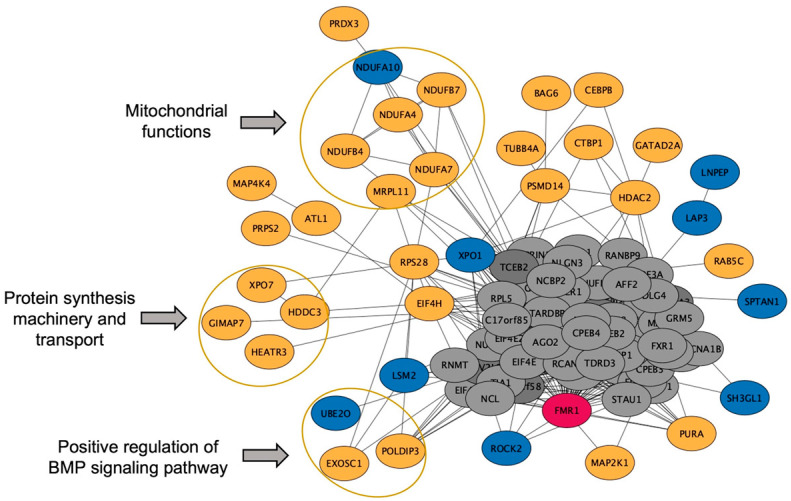
Molecular functions altered between CON and NCON at V1 and V2. Gene ontology molecular functions, including the mitochondrial, protein synthesis machinery and transport, and positive regulation of BMP signaling pathways enriched in the proteins (enclosed in orange circle) that are consistently differentially expressed between CON and NCON at V1 and V2. The blue color represents the proteins that are up-regulated in CON. Yellow represents the down-regulated ones. While Grey is representing the non-differential proteins and Red is *FMR1*.

**Figure 7 ijms-24-13477-f007:**
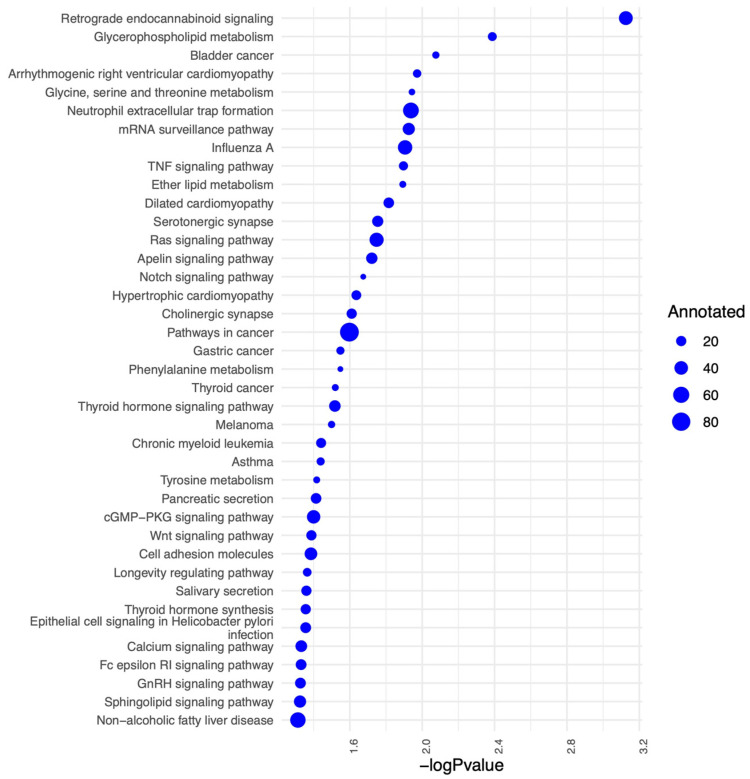
Significantly altered pathways comparing CON to NCON at V2. Protein-protein interactions from STRING database are represented as edges between proteins.

**Figure 8 ijms-24-13477-f008:**
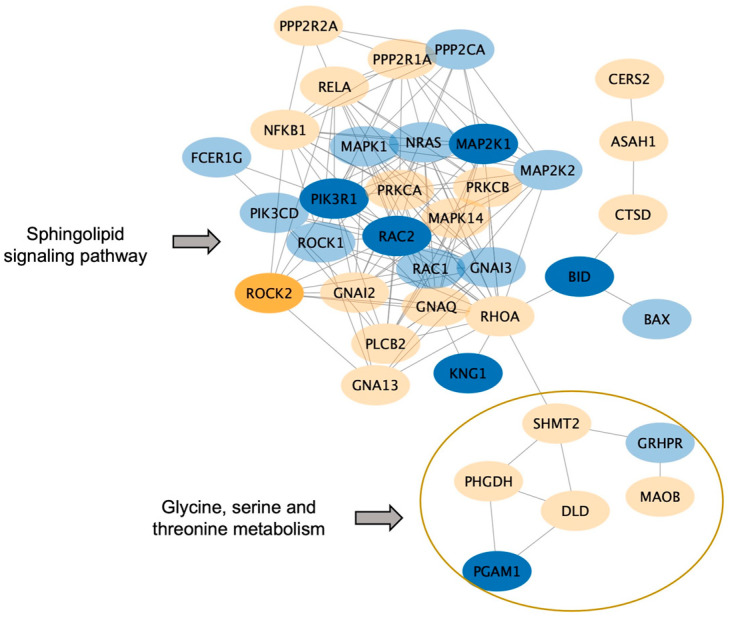
Sphingolipid and amino acid metabolism altered in CON. Proteins associated with the sphingolipid and amino-acid pathways, including glycine, serine, and threonine metabolism (enclosed in orange circle), are found to be enriched in the comparison between CON and NCON at V2. The blue color represents the proteins that are up-regulated in CON. Yellow represents the down-regulated ones. The level of the significance is indicated with the intensity of the color.

**Figure 9 ijms-24-13477-f009:**
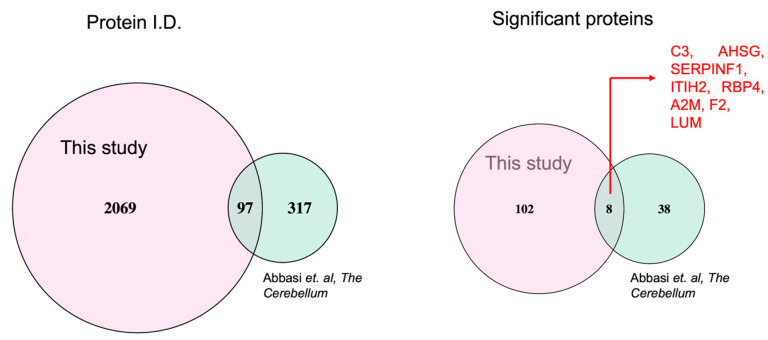
Comparison of CSF and blood proteomic profile. Cerebrospinal fluid (CSF) proteome of FXTAS patients identified 414 [13]. Blood proteome profile of CON at V2 (in pink) identified a total of 2166 proteins of which 97 were found in common. By looking at significantly altered proteins from blood proteomic profile (n = 110) and CSF proteomic profile (n = 46), 8 proteins were found to be in common (indicated in red ink), including Complement C3, Alpha-2-HS-glycoprotein, Pigment epithelium-derived factor, Inter-alpha-trypsin inhibitor heavy chain H2, Retinol-binding protein 4, Alpha-2-macroglobulin, Prothrombin, and Lumican.

**Table 1 ijms-24-13477-t001:** Subjects Baseline Characteristics.

	Non-Converter (n = 19)	Converter (n = 17)	Healthy Control (n = 12)	*p*-Value
**Age**				
N	19	17	12	
Mean (SD)	57.2 (8.2)	53.2 (6.9)	50.2 (5.2)	0.0319
Median (Range)	59 (44–68)	53 (42–65)	49 (45–63)	
**CGG repeat**				
N	19	17	12	
Mean (SD)	82.9 (22)	90.2 (21.4)	29.8 (2.4)	<0.001
Median (Range)	78 (56–135)	85 (60–141)	30 (23–32)	

**Table 2 ijms-24-13477-t002:** Differential expression statistics (BH adjusted *p*-values) among converters, non-converters, and healthy controls.

Sr #	PG Protein Accessions	PG Genes	PG Protein Descriptions	Converter V2_v_HealthyControl	Converter_v_NonConverter_V1	Converter_v_NonConverter_V2	Pre_v_Control_Baseline	V2_v_V1_Converter	V2_v_V1_NonConverter
1	P55957	BID	BH3-interacting domain death agonist	0.330961	0.000163	0.0184426	0.903113	0.000126	0.2063338
2	P00403	MT-CO2	Cytochrome c oxidase subunit 2	0.001902	0.0000083	0.2557524	0.386688	8.5 × 10^−6^	0.5831517
3	O75531	BANF1	Barrier-to-autointegration factor	0.0001968	0.0000173	0.1435231	0.176737	0.003215	0.9627627
4	P20674	COX5A	Cytochrome c oxidase subunit 5A, mitochondrial	0.0001968	0.0000002	0.7346092	0.13	5 × 10^−7^	0.9955487
5	Q8NFW8	CMAS	N-acylneuraminate cytidylyltransferase	0.707128	0.000004	0.0590233	0.958116	0.000762	0.8246921
6	Q15370	ELOB	Elongin-B	0.9778348	0.0005851	0.0004076	0.886594	0.234448	0.499658
7	Q9Y3B2	EXOSC1	Exosome complex component CSL4	0.9398573	0.0000276	0.0450991	0.91006	0.009782	0.8639569
8	P62310	LSM3	U6 snRNA-associated Sm-like protein LSm3	0.0135975	0.0000349	0.4197721	0.342944	9.16 × 10^−5^	0.8092159
9	Q92769	HDAC2	Histone deacetylase 2	0.5817782	0.0015861	0.002199	0.421168	0.055764	0.3596959
10	P42025	ACTR1B	Beta-centractin	0.7463896	0.0034287	0.0000198	0.920365	0.498285	0.2063338
11	P17676	CEBPB	CCAAT/enhancer-binding protein beta	0.3772653	0.0006946	0.0443946	0.649723	0.035408	0.7433883
12	Q6P1A2	LPCAT3	Lysophospholipid acyltransferase 5	0.2781298	0.0015861	0.077895	0.783158	0.008247	0.499658
13	O00422	SAP18	Histone deacetylase complex subunit SAP18	0.2733554	0.0082336	0.1201545	0.941093	0.003919	0.3596959
14	P14406	COX7A2	Cytochrome c oxidase subunit 7A2, mitochondrial	0.0793535	0.0017415	0.1493414	0.477268	0.008643	0.6176757
15	P01909	HLA-DQA1	HLA class II histocompatibility antigen, DQ alpha 1 chain	0.8705704	0.0016708	0.0405223	0.389286	0.014566	0.4822303
16	O95716	RAB3D	Ras-related protein Rab-3D	0.4995509	0.0015529	0.0639465	0.833119	0.031063	0.6751625
17	O95182	NDUFA7	NADH dehydrogenase [ubiquinone] 1 alpha subcomplex subunit 7	0.8813178	0.0191425	0.0030663	0.625891	0.114989	0.2063338
18	P83881; Q969Q0	RPL36A; RPL36AL	60S ribosomal protein L36a; 60S ribosomal protein L36a-like	0.3613339	0.0037113	0.132179	0.890401	0.033957	0.6898521
19	Q8N699	MYCT1	Myc target protein 1	0.5533207	0.0043394	0.0155536	0.942197	0.017778	0.2948038
20	P51148	RAB5C	Ras-related protein Rab-5C	0.4621168	0.003153	0.0386719	0.630496	0.088983	0.6539073
21	P12829	MYL4	Myosin light chain 4	0.8810339	0.0180141	0.0002622	0.937117	0.633167	0.2948038
22	Q8NHV1	GIMAP7	GTPase IMAP family member 7	0.5083015	0.0011011	0.0041675	0.585434	0.14075	0.5669873
23	Q9UDW1	UQCR10	Cytochrome b-c1 complex subunit 9	0.6214831	0.0185665	0.1445605	0.771541	0.010801	0.4096213
24	P62854	RPS26	40S ribosomal protein S26	0.7675159	0.0011011	0.1690844	0.846943	0.008895	0.7317184
25	P07919	UQCRH	Cytochrome b-c1 complex subunit 6, mitochondrial	0.0276532	0.0006946	0.7931993	0.688736	0.000646	0.9420505
26	Q96CN7	ISOC1	Isochorismatase domain-containing protein 1	0.9540223	0.0006946	0.2221959	0.642702	0.004467	0.7459911
27	P30048	PRDX3	Thioredoxin-dependent peroxide reductase, mitochondrial	0.8126683	0.0024886	0.0008139	0.903113	0.583097	0.633638
28	O14980	XPO1	Exportin-1	0.496806	0.0033645	0.0157841	0.292174	0.088983	0.5036518
29	P10155	RO60	60 kDa SS-A/Ro ribonucleoprotein	0.8958494	0.003992	0.0982565	0.879425	0.041899	0.6641875
30	P14854	COX6B1	Cytochrome c oxidase subunit 6B1	0.2523229	0.0019848	0.2434441	0.692178	0.015693	0.8107049
31	P27338	MAOB	Amine oxidase [flavin-containing] B	0.7579699	0.0114213	0.0791405	0.958401	0.077641	0.5831517
32	Q9NWH9	SLTM	SAFB-like transcription modulator	0.65767	0.0011011	0.3615714	0.822037	0.00603	0.8663002
33	Q7LBR1	CHMP1B	Charged multivesicular body protein 1b	0.9983177	0.0080553	0.077895	0.757183	0.048739	0.5340208
34	O14949	UQCRQ	Cytochrome b-c1 complex subunit 8	0.2781298	0.0025569	0.3618405	0.864467	0.004467	0.7117886
35	O76021	RSL1D1	Ribosomal L1 domain-containing protein 1	0.9556111	0.0029101	0.0961799	0.934403	0.082046	0.8256684
36	Q9BY77	POLDIP3	Polymerase delta-interacting protein 3	0.9939751	0.0015861	0.0193043	0.971953	0.18299	0.7712084
37	P84098	RPL19	60S ribosomal protein L19	0.447567	0.0026024	0.208154	0.64276	0.050668	0.9332484
38	Q8NEW0	SLC30A7	Zinc transporter 7	0.9047438	0.0011011	0.2681998	0.785538	0.014566	0.9132308
39	O60831	PRAF2	PRA1 family protein 2	0.7675159	0.0187209	0.0445152	0.921153	0.191474	0.5792395
40	P25490	YY1	Transcriptional repressor protein YY1	0.3425898	0.0006946	0.1971043	0.340162	0.108937	0.7852614
41	P54578	USP14	Ubiquitin carboxyl-terminal hydrolase 14	0.7001369	0.0117166	0.155195	0.914396	0.011072	0.4529446
42	O95299	NDUFA10	NADH dehydrogenase [ubiquinone] 1 alpha subcomplex subunit 10, mitochondrial	0.7675159	0.02368	0.0177951	0.251517	0.027293	0.2063338
43	P50552	VASP	Vasodilator-stimulated phosphoprotein	0.0001968	0.0006946	0.1645395	0.281154	1.93 × 10^−5^	0.7253255
44	P62993	GRB2	Growth factor receptor-bound protein 2	0.0002878	0.0206862	0.9579479	0.11054	0.003685	0.8107049
45	Q9NRX4	PHPT1	14 kDa phosphohistidine phosphatase	0.6110357	0.0047335	0.2458114	0.881767	0.075083	0.9420505
46	P51809	VAMP7	Vesicle-associated membrane protein 7	0.9487359	0.1044092	0.0007722	0.980205	0.314721	0.0612594
47	Q9Y3A3	MOB4	MOB-like protein phocein	0.5092093	0.0915321	0.0523055	0.780941	0.251271	0.4529446
48	Q9P1F3	ABRACL	Costars family protein ABRACL	0.8953855	0.0308248	0.1269368	0.773045	0.048739	0.481216
49	P30049	ATP5F1D	ATP synthase subunit delta, mitochondrial	0.0827237	0.0242103	0.1852471	0.229557	0.178417	0.7992426
50	P42285	MTREX	Exosome RNA helicase MTR4	0.4153117	0.0080553	0.0122675	0.421857	0.194376	0.499658
51	Q9UK76	JPT1	Jupiter microtubule associated homolog 1	0.5063727	0.0189857	0.1924087	0.831264	0.04887	0.6224493
52	Q8IV08	PLD3	Phospholipase D3	0.8688999	0.0030172	0.0430464	0.783158	0.092923	0.6898521
53	P62306	SNRPF	Small nuclear ribonucleoprotein F	0.0320778	0.0117166	0.7274526	0.752206	0.010346	0.8387102
54	Q99536	VAT1	Synaptic vesicle membrane protein VAT-1 homolog	0.3425898	0.0029101	0.3914526	0.60927	0.019668	0.9275825
55	Q15102	PAFAH1B3	Platelet-activating factor acetylhydrolase IB subunit gamma	0.7602693	0.0043843	0.3073004	0.833119	0.011757	0.7161634
56	Q9Y3B7	MRPL11	39S ribosomal protein L11, mitochondrial	0.4919452	0.0167599	0.0333048	0.30862	0.096942	0.4594807
57	Q9Y5Z4	HEBP2	Heme-binding protein 2	0.8966822	0.0242103	0.0410507	0.950572	0.106176	0.4529446
58	O95139	NDUFB6	NADH dehydrogenase [ubiquinone] 1 beta subcomplex subunit 6	0.7001369	0.0109981	0.2656104	0.881767	0.030425	0.7063057
59	Q6IAA8	LAMTOR1	Ragulator complex protein LAMTOR1	0.8892256	0.0676655	0.0333048	0.553347	0.137919	0.3596959
60	O00483	NDUFA4	Cytochrome c oxidase subunit NDUFA4	0.7787789	0.0043843	0.0004076	0.771541	0.968941	0.6960786
61	P49065	ALB	Serum albumin	0.0001968	0.3958862	0.0017581	0.101752	0.050668	0.9332484
62	P04217	A1BG	Alpha-1B-glycoprotein	0.0000005	0.0882604	0.0008139	0.076083	0.310441	0.8111847
63	P02452	COL1A1	Collagen alpha-1(I) chain	0.0001968	0.9098129	0.0112894	0.012023	0.020765	0.8570423
64	P10412	HIST1H1E	Histone H1.4	0.0002878	0.1925862	0.0030893	0.058588	0.006362	0.4096213
65	P02760	AMBP	Protein AMBP	0.0002878	0.2220095	0.0091412	0.058588	0.422431	0.8428062
66	P05543	SERPINA7	Thyroxine-binding globulin	0.0056139	0.3415665	0.0014646	0.23407	0.12332	0.7834307
67	P60660	MYL6	Myosin light polypeptide 6	0.0002878	0.3674064	0.0199639	0.024347	0.041899	0.499658
68	P08697	SERPINF2	Alpha-2-antiplasmin	0.0007936	0.0297095	0.0004076	0.237164	0.414601	0.8286978
69	Q15907	RAB11B	Ras-related protein Rab-11B	0.0024077	0.7777295	0.0092849	0.38221	0.011072	0.8428062
70	Q9Y6W5	WASF2	Wiskott-Aldrich syndrome protein family member 2	0.0002878	0.4113262	0.0073602	0.468022	0.004467	0.7106443
71	P12109	COL6A1	Collagen alpha-1(VI) chain	0.001902	0.8530394	0.0200518	0.292934	0.029598	0.9623254
72	Q9BRA2	TXNDC17	Thioredoxin domain-containing protein 17	0.0045339	0.8419097	0.020576	0.278214	0.115543	0.8520701
73	Q8N386	LRRC25	Leucine-rich repeat-containing protein 25	0.0349695	0.0514834	0.000297	0.433359	0.574902	0.5565245
74	P51884	LUM	Lumican	0.0119658	0.502447	0.0184426	0.30862	0.27929	0.8465036
75	O95168	NDUFB4	NADH dehydrogenase [ubiquinone] 1 beta subcomplex subunit 4	0.005601	0.0199489	0.0003307	0.32246	0.929963	0.5115525
76	P63027	VAMP2	Vesicle-associated membrane protein 2	0.0003153	0.956101	0.3489073	0.068855	0.291336	0.9204359
77	Q8IZ07	ANKRD13A	Ankyrin repeat domain-containing protein 13A	0.0912438	0.1693605	0.0021475	0.147105	0.80839	0.4980031
78	Q8WYJ6	SEPTIN1	Septin-1	0.0272654	0.6266639	0.0882902	0.358434	0.152744	0.876325
79	P51570	GALK1	Galactokinase	0.0171924	0.5636075	0.0008139	0.183522	0.658335	0.2746342

**Table 3 ijms-24-13477-t003:** Differentially expressed proteins between converter and non-converter at Visit 1.

Sr #	Protein Accessions	Genes	logFC	Ave Expr	*p*-Value	adj.P Val	Protein Descriptions
1	P20674	COX5A	1.11	13.73	8.45 × 10^−11^	1.83 × 10^−7^	Cytochrome c oxidase subunit 5A, mitochondrial
2	Q8NFW8	CMAS	1.82	13.4	3.73 × 10^−9^	4.04 × 10^−6^	N-acylneuraminate cytidylyltransferase
3	P00403	MT-CO2	1.22	14.45	1.15 × 10^−8^	8.27 × 10^−6^	Cytochrome c oxidase subunit 2
4	O75531	BANF1	3.23	13.92	3.19 × 10^−8^	1.73 × 10^−5^	Barrier-to-autointegration factor
5	Q9Y3B2	EXOSC1	0.85	12.98	6.37 × 10^−8^	2.76 × 10^−5^	Exosome complex component CSL4
6	P62310	LSM3	1.09	13.49	9.67 × 10^−8^	3.49 × 10^−5^	U6 snRNA-associated Sm-like protein LSm3
7	P55957	BID	1.59	13.87	5.27 × 10^−7^	0.00016297	BH3-interacting domain death agonist
8	Q15370	ELOB	1	13.11	2.16 × 10^−6^	0.00058509	Elongin-B
9	P25490	YY1	0.69	13.27	3.26 × 10^−6^	0.0006946	Transcriptional repressor protein YY1
10	Q96N66	MBOAT7	1.49	13.45	3.48 × 10^−6^	0.0006946	Lysophospholipid acyltransferase 7
11	P17676	CEBPB	2.16	13.57	3.67 × 10^−6^	0.0006946	CCAAT/enhancer-binding protein beta
12	P50552	VASP	−0.45	15.4	3.90 × 10^−6^	0.0006946	Vasodilator-stimulated phosphoprotein
13	Q96CN7	ISOC1	1.29	13.18	4.39 × 10^−6^	0.0006946	Isochorismatase domain-containing protein 1
14	P07919	UQCRH	0.97	13.2	4.49 × 10^−6^	0.0006946	Cytochrome b-c1 complex subunit 6, mitochondrial
15	P62854	RPS26	2.28	14.52	8.22 × 10^−6^	0.00110109	40S ribosomal protein S26
16	Q8NHV1	GIMAP7	0.9	13	8.71 × 10^−6^	0.00110109	GTPase IMAP family member 7
17	Q8NEW0	SLC30A7	2.29	13.65	8.81 × 10^−6^	0.00110109	Zinc transporter 7
18	Q9NWH9	SLTM	3.08	13.68	9.48 × 10^−6^	0.00110109	SAFB-like transcription modulator
19	O15347	HMGB3	0.84	15.94	9.66 × 10^−6^	0.00110109	High mobility group protein B3
20	O95716	RAB3D	2.06	14.75	1.43 × 10^−5^	0.00155291	Ras-related protein Rab-3D
21	Q92769	HDAC2	0.52	12.95	1.58 × 10^−5^	0.00158606	Histone deacetylase 2
22	P31949	S100A11	0.77	14.52	1.66 × 10^−5^	0.00158606	Protein S100-A11
23	Q9BY77	POLDIP3	0.87	12.95	1.75 × 10^−5^	0.00158606	Polymerase delta-interacting protein 3
24	Q6P1A2	LPCAT3	1.52	13.45	1.76 × 10^−5^	0.00158606	Lysophospholipid acyltransferase 5
25	P01909	HLA-DQA1	0.46	12.9	1.93 × 10^−5^	0.00167082	HLA class II histocompatibility antigen, DQ alpha 1 chain
26	P14406	COX7A2	1.56	13.67	2.09 × 10^−5^	0.00174149	Cytochrome c oxidase subunit 7A2, mitochondrial
27	P14854	COX6B1	1.13	13.72	2.47 × 10^−5^	0.00198483	Cytochrome c oxidase subunit 6B1
28	Q9UBW5	BIN2	−0.33	14.52	2.93 × 10^−5^	0.00226447	Bridging integrator 2
29	P02656	APOC3	1.58	14.74	3.27 × 10^−5^	0.00244275	Apolipoprotein C-III
30	P30048	PRDX3	0.75	14.71	3.45 × 10^−5^	0.00248864	Thioredoxin-dependent peroxide reductase, mitochondrial
31	P62857	RPS28	0.99	15.39	3.69 × 10^−5^	0.00255693	40S ribosomal protein S28
32	O14949	UQCRQ	1.18	13.37	3.78 × 10^−5^	0.00255693	Cytochrome b-c1 complex subunit 8
33	P84098	RPL19	1.21	14.21	3.96 × 10^−5^	0.00260242	60S ribosomal protein L19
34	O43760	SYNGR2	0.8	13.78	4.20 × 10^−5^	0.00267673	Synaptogyrin-2
35	Q02750	MAP2K1	2.54	15.39	4.77 × 10^−5^	0.00291015	Dual specificity mitogen-activated protein kinase kinase 1
36	Q99536	VAT1	0.53	13.63	4.91 × 10^−5^	0.00291015	Synaptic vesicle membrane protein VAT-1 homolog
37	O76021	RSL1D1	1.07	13.43	5.09 × 10^−5^	0.00291015	Ribosomal L1 domain-containing protein 1
38	P62995	TRA2B	1.26	13.66	5.11 × 10^−5^	0.00291015	Transformer-2 protein homolog beta
39	Q8IV08	PLD3	0.63	12.92	5.48 × 10^−5^	0.00301717	Phospholipase D3
40	P98179	RBM3	0.68	13.43	5.57 × 10^−5^	0.00301717	RNA-binding protein 3
41	P51148	RAB5C	0.86	13.94	5.97 × 10^−5^	0.00315297	Ras-related protein Rab-5C
42	O14980	XPO1	−0.26	12.95	6.52 × 10^−5^	0.00336451	Exportin-1
43	Q02108	GUCY1A1	0.49	12.9	6.71 × 10^−5^	0.00338228	Guanylate cyclase soluble subunit alpha-1
44	O75439	PMPCB	1.38	13.85	7.10 × 10^−5^	0.00342866	Mitochondrial-processing peptidase subunit beta
45	Q7Z4Q2	HEATR3	0.44	12.81	7.36 × 10^−5^	0.00342866	HEAT repeat-containing protein 3
46	Q13884	SNTB1	−0.31	13.21	7.37 × 10^−5^	0.00342866	Beta-1-syntrophin
47	Q9Y266	NUDC	1.4	13.07	7.57 × 10^−5^	0.00342866	Nuclear migration protein nudC
48	P42025	ACTR1B	0.52	12.99	7.60 × 10^−5^	0.00342866	Beta-centractin
49	Q04323	UBXN1	2.36	13.61	8.12 × 10^−5^	0.00359083	UBX domain-containing protein 1
50	P83881; Q969Q0	RPL36A; RPL36AL	1.24	13.66	8.57 × 10^−5^	0.00371127	60S ribosomal protein L36a; 60S ribosomal protein L36a-like
51	Q86WV1	SKAP1	1.57	13.51	9.38 × 10^−5^	0.00398485	Src kinase-associated phosphoprotein 1
52	P10155	RO60	0.51	13.52	9.58 × 10^−5^	0.00399203	60 kDa SS-A/Ro ribonucleoprotein
53	P62877	RBX1	3.27	14.68	0.00010525	0.00428208	E3 ubiquitin-protein ligase RBX1
54	P53041	PPP5C	0.93	13.06	0.00010676	0.00428208	Serine/threonine-protein phosphatase 5
55	Q8N699	MYCT1	1.31	13.32	0.00011019	0.00433936	Myc target protein 1
56	Q15102	PAFAH1B3	1.69	13.31	0.00011444	0.00438431	Platelet-activating factor acetylhydrolase IB subunit gamma
57	O00483	NDUFA4	1.95	14.82	0.00011538	0.00438431	Cytochrome c oxidase subunit NDUFA4
58	Q86YP4	GATAD2A	2.45	14.08	0.000118	0.00440654	Transcriptional repressor p66-alpha
59	Q9NRX4	PHPT1	0.63	12.9	0.0001294	0.00473352	14 kDa phosphohistidine phosphatase
60	O75116	ROCK2	−0.27	13.28	0.00013112	0.00473352	Rho-associated protein kinase 2
61	P55265	ADAR	−0.36	13.04	0.00013491	0.00479042	Double-stranded RNA-specific adenosine deaminase
62	P16333	NCK1	1.05	13.17	0.00013831	0.00480724	Cytoplasmic protein NCK1
63	P31645	SLC6A4	1.57	13.26	0.00013982	0.00480724	Sodium-dependent serotonin transporter
64	Q9UK45	LSM7	4.39	15.02	0.00014412	0.00486792	U6 snRNA-associated Sm-like protein LSm7
65	P0DP23; P0DP24; P0DP25	CALM1; CALM2; CALM3	0.49	13.84	0.00014608	0.00486792	Calmodulin-1; Calmodulin-2; Calmodulin-3
66	P78406	RAE1	−0.36	13.07	0.00014953	0.00490735	mRNA export factor
67	O95433	AHSA1	0.99	13.17	0.0001634	0.00528231	Activator of 90 kDa heat shock protein ATPase homolog 1
68	Q9BQ61	TRIR	0.78	12.89	0.00016598	0.00528682	Telomerase RNA component interacting RNase
69	P04350	TUBB4A	0.56	12.94	0.00017133	0.0053782	Tubulin beta-4A chain
70	P02751	FN1	−0.3	13.7	0.0002119	0.00655692	Fibronectin
71	Q13363	CTBP1	0.42	13.34	0.00025616	0.00781477	C-terminal-binding protein 1
72	Q7LBR1	CHMP1B	2.32	13.74	0.00027002	0.00805526	Charged multivesicular body protein 1b
73	P42285	MTREX	0.52	12.85	0.00027148	0.00805526	Exosome RNA helicase MTR4
74	O00422	SAP18	2.5	14.61	0.00028182	0.00823363	Histone deacetylase complex subunit SAP18
75	O00193	SMAP	0.43	12.87	0.0002851	0.00823363	Small acidic protein
76	Q9Y5X3	SNX5	−0.56	12.78	0.00036723	0.01046605	Sorting nexin-5
77	P20339	RAB5A	0.71	14.8	0.00037375	0.01051357	Ras-related protein Rab-5A
78	P62273	RPS29	0.52	13.31	0.00038065	0.01057044	40S ribosomal protein S29
79	P46379	BAG6	0.91	13.18	0.00040004	0.01096815	Large proline-rich protein BAG6
80	O95139	NDUFB6	0.99	13.25	0.00040621	0.01099807	NADH dehydrogenase [ubiquinone] 1 beta subcomplex subunit 6
81	P63165	SUMO1	2	13.75	0.00042395	0.01133665	Small ubiquitin-related modifier 1
82	P27338	MAOB	−0.3	13.33	0.00043239	0.01142134	Amine oxidase [flavin-containing] B
83	Q9H4G4	GLIPR2	0.54	15.52	0.00045783	0.0117166	Golgi-associated plant pathogenesis-related protein 1
84	Q96EY8	MMAB	0.47	12.96	0.00045894	0.0117166	Corrinoid adenosyltransferase
85	O00487	PSMD14	1.12	13.34	0.00046704	0.0117166	26S proteasome non-ATPase regulatory subunit 14
86	P04424	ASL	0.45	13.02	0.00046985	0.0117166	Argininosuccinate lyase
87	Q96K37	SLC35E1	−0.44	12.86	0.00047521	0.0117166	Solute carrier family 35 member E1
88	A5YKK6	CNOT1	0.46	12.8	0.00047814	0.0117166	CCR4-NOT transcription complex subunit 1
89	P62306	SNRPF	1.14	14.31	0.00048471	0.0117166	Small nuclear ribonucleoprotein F
90	P54578	USP14	0.51	13.49	0.00048684	0.0117166	Ubiquitin carboxyl-terminal hydrolase 14
91	Q9UIA9	XPO7	2.14	14.44	0.00050346	0.01188018	Exportin-7
92	P49959	MRE11	−0.3	12.9	0.00050461	0.01188018	Double-strand break repair protein MRE11
93	P18206	VCL	−0.32	14.87	0.0005265	0.01226224	Vinculin
94	O43290	SART1	1.32	13.35	0.00054598	0.01258082	U4/U6.U5 tri-snRNP-associated protein 1
95	O95819	MAP4K4	0.57	14.63	0.00059275	0.01351476	Mitogen-activated protein kinase kinase kinase kinase 4
96	Q5T1M5	FKBP15	0.48	13.35	0.00060115	0.01356334	FK506-binding protein 15
97	P02765	AHSG	−0.74	18.91	0.00061844	0.0137706	Alpha-2-HS-glycoprotein
98	O00170	AIP	−0.35	13.19	0.00062305	0.0137706	AH receptor-interacting protein
99	Q12907	LMAN2	0.58	14.57	0.0006296	0.01377496	Vesicular integral-membrane protein VIP36
100	Q8NFV4	ABHD11	0.39	12.85	0.00065616	0.01421252	Protein ABHD11
101	P63000	RAC1	−0.26	15.49	0.00066636	0.01429044	Ras-related C3 botulinum toxin substrate 1
102	Q8NCG7	DAGLB	0.52	12.87	0.000727	0.01543803	Sn1-specific diacylglycerol lipase beta
103	Q9BQE9	BCL7B	2.06	13.52	0.00078504	0.0165087	B-cell CLL/lymphoma 7 protein family member B
104	P61964	WDR5	1.58	13.48	0.00080743	0.01675991	WD repeat-containing protein 5
105	Q9Y3B7	MRPL11	0.83	13.36	0.00081246	0.01675991	39S ribosomal protein L11, mitochondrial
106	Q8IX12	CCAR1	0.87	13.45	0.00084694	0.01730627	Cell division cycle and apoptosis regulator protein 1
107	Q01658	DR1	1.6	13.34	0.0008722	0.01764242	Protein Dr1
108	Q8IVB4	SLC9A9	0.72	13.01	0.00087968	0.01764242	Sodium/hydrogen exchanger 9
109	P12829	MYL4	1.01	13.41	0.00090653	0.01801412	Myosin light chain 4
110	Q99961	SH3GL1	−0.34	13.41	0.00092381	0.01819059	Endophilin-A2
111	Q9UDW1	UQCR10	0.46	12.95	0.00095147	0.01856648	Cytochrome b-c1 complex subunit 9
112	O60831	PRAF2	0.79	13.17	0.00096803	0.01872093	PRA1 family protein 2
113	Q9UK76	JPT1	0.36	13.01	0.00099048	0.01898573	Jupiter microtubule associated homolog 1
114	P41226	UBA7	−0.28	13.15	0.00101087	0.01910907	Ubiquitin-like modifier-activating enzyme 7
115	O15173	PGRMC2	0.3	13.43	0.00101456	0.01910907	Membrane-associated progesterone receptor component 2
116	O95182	NDUFA7	1.09	13.78	0.00102517	0.01914249	NADH dehydrogenase [ubiquinone] 1 alpha subcomplex subunit 7
117	P62633	CNBP	0.87	13.43	0.00104734	0.01928702	Cellular nucleic acid-binding protein
118	Q9UIG0	BAZ1B	1.4	13.31	0.00105072	0.01928702	Tyrosine-protein kinase BAZ1B
119	O95168	NDUFB4	1.4	14.47	0.00109599	0.01994894	NADH dehydrogenase [ubiquinone] 1 beta subcomplex subunit 4
120	Q93008	USP9X	−0.37	13.01	0.001115	0.02012574	Probable ubiquitin carboxyl-terminal hydrolase FAF-X
121	Q8WXF1	PSPC1	0.43	13.62	0.0011414	0.02040483	Paraspeckle component 1
122	Q96JB5	CDK5RAP3	1.49	13.69	0.0011493	0.02040483	CDK5 regulatory subunit-associated protein 3
123	Q6DD87	ZNF787	0.45	12.84	0.00116082	0.02044178	Zinc finger protein 787
124	Q7Z6Z7	HUWE1	−0.24	12.95	0.00117746	0.02056758	E3 ubiquitin-protein ligase HUWE1
125	P62993	GRB2	0.35	14.04	0.0011938	0.02068617	Growth factor receptor-bound protein 2
126	P68402	PAFAH1B2	0.44	12.91	0.00122782	0.0210104	Platelet-activating factor acetylhydrolase IB subunit beta
127	Q7RTV0	PHF5A	1.95	14.44	0.00123191	0.0210104	PHD finger-like domain-containing protein 5A
128	O75368	SH3BGRL	0.58	14.02	0.00125309	0.02120469	SH3 domain-binding glutamic acid-rich-like protein
129	Q9Y4L1	HYOU1	−0.25	13.33	0.00127233	0.02136329	Hypoxia up-regulated protein 1
130	Q9P035	HACD3	−0.29	13.02	0.00128789	0.02145828	Very-long-chain (3R)-3-hydroxyacyl-CoA dehydratase 3
131	Q9Y333	LSM2	−0.33	12.85	0.0013131	0.0215621	U6 snRNA-associated Sm-like protein LSm2
132	Q8TBC4	UBA3	1.97	13.45	0.00132373	0.0215621	NEDD8-activating enzyme E1 catalytic subunit
133	Q9C0C9	UBE2O	−0.27	13.08	0.00132399	0.0215621	(E3-independent) E2 ubiquitin-conjugating enzyme
134	Q92542	NCSTN	1.03	13.26	0.00147097	0.02366738	Nicastrin
135	Q15056	EIF4H	0.62	13.81	0.00147511	0.02366738	Eukaryotic translation initiation factor 4H
136	O95299	NDUFA10	−0.22	12.9	0.00148683	0.02368002	NADH dehydrogenase [ubiquinone] 1 alpha subcomplex subunit 10, mitochondrial
137	P30049	ATP5F1D	1.96	15.01	0.0015432	0.02421028	ATP synthase subunit delta, mitochondrial
138	Q9H3P7	ACBD3	0.94	13.22	0.00154752	0.02421028	Golgi resident protein GCP60
139	Q96EP5	DAZAP1	0.78	14.45	0.00155565	0.02421028	DAZ-associated protein 1
140	O00299	CLIC1	−0.26	16.01	0.00157157	0.02421028	Chloride intracellular channel protein 1
141	Q9Y5Z4	HEBP2	0.23	13.05	0.00157602	0.02421028	Heme-binding protein 2
142	P49585	PCYT1A	0.62	13.12	0.00158822	0.02422601	Choline-phosphate cytidylyltransferase A
143	P19784	CSNK2A2	1.44	13.1	0.0016445	0.024909	Casein kinase II subunit alpha’
144	A0AVT1	UBA6	−0.26	12.99	0.00168755	0.02533419	Ubiquitin-like modifier-activating enzyme 6
145	P23141	CES1	0.38	13.73	0.00169596	0.02533419	Liver carboxylesterase 1
146	P21333	FLNA	−0.28	15.69	0.0017293	0.02565525	Filamin-A
147	P04075	ALDOA	−0.25	15.31	0.00186956	0.02749417	Fructose-bisphosphate aldolase A
148	Q8IZP0	ABI1	0.28	12.97	0.00187864	0.02749417	Abl interactor 1
149	Q9C0E8	LNPK	0.67	13.08	0.00194398	0.02818302	Endoplasmic reticulum junction formation protein lunapark
150	P00747	PLG	−0.35	13.73	0.00195293	0.02818302	Plasminogen
151	P06239	LCK	0.42	13.25	0.00196621	0.02818302	Tyrosine-protein kinase Lck
152	O14735	CDIPT	1.11	13.53	0.00197883	0.02818302	CDP-diacylglycerol-inositol 3-phosphatidyltransferase
153	O00154	ACOT7	−0.35	12.91	0.00199077	0.02818302	Cytosolic acyl coenzyme A thioester hydrolase
154	Q13435	SF3B2	−0.27	13.78	0.00200778	0.02823925	Splicing factor 3B subunit 2
155	Q14642	INPP5A	0.79	13.03	0.00202716	0.02828249	Inositol polyphosphate-5-phosphatase A
156	P62328	TMSB4X	−0.51	14.4	0.00203697	0.02828249	Thymosin beta-4
157	Q9Y3Y2	CHTOP	0.66	13.88	0.00206287	0.02830664	Chromatin target of PRMT1 protein
158	Q8NBQ5	HSD17B11	0.58	13.12	0.00206484	0.02830664	Estradiol 17-beta-dehydrogenase 11
159	Q15833	STXBP2	−0.21	13.63	0.00209062	0.0284634	Syntaxin-binding protein 2
160	Q32P28	P3H1	0.46	12.87	0.00210256	0.0284634	Prolyl 3-hydroxylase 1
161	Q9Y5S9	RBM8A	0.46	14.02	0.00214639	0.02887632	RNA-binding protein 8A
162	Q8N392	ARHGAP18	−0.36	13.81	0.00217155	0.02903448	Rho GTPase-activating protein 18
163	P04234	CD3D	1.07	13.67	0.00221957	0.02942847	T-cell surface glycoprotein CD3 delta chain
164	P61803	DAD1	0.97	14.32	0.00223253	0.02942847	Dolichyl-diphosphooligosaccharide-protein glycosyltransferase subunit DAD1
165	Q01518	CAP1	−0.21	15.06	0.00224178	0.02942847	Adenylyl cyclase-associated protein 1
166	P08697	SERPINF2	1.26	15.88	0.0022769	0.02970948	Alpha-2-antiplasmin
167	Q8WXF7	ATL1	0.35	12.87	0.00233705	0.03031163	Atlastin-1
168	Q6P1M0	SLC27A4	0.61	12.78	0.00235843	0.03040693	Long-chain fatty acid transport protein 4
169	Q9P1F3	ABRACL	−0.2	12.79	0.00240918	0.03082475	Costars family protein ABRACL
170	P01911	HLA-DRB1	0.64	14.51	0.0024211	0.03082475	HLA class II histocompatibility antigen, DRB1-15 beta chain
171	Q16643	DBN1	−0.3	13.49	0.00243353	0.03082475	Drebrin
172	P11908	PRPS2	1.43	13.64	0.00245004	0.03085338	Ribose-phosphate pyrophosphokinase 2
173	Q99439	CNN2	−0.29	14.25	0.00250122	0.03131584	Calponin-2
174	Q86UT6	NLRX1	−0.35	12.84	0.00251982	0.03134018	NLR family member X1
175	Q9NRL3	STRN4	1.43	14.38	0.0025321	0.03134018	Striatin-4
176	Q9NZ45	CISD1	−0.42	12.92	0.00257935	0.03174358	CDGSH iron-sulfur domain-containing protein 1
177	P84103	SRSF3	0.54	13.92	0.00270148	0.03305881	Serine/arginine-rich splicing factor 3
178	Q8TF42	UBASH3B	−0.31	13.22	0.00284578	0.03451842	Ubiquitin-associated and SH3 domain-containing protein B
179	P98194	ATP2C1	−0.35	13.11	0.00285263	0.03451842	Calcium-transporting ATPase type 2C member 1
180	O95870	ABHD16A	−0.22	13.1	0.00291559	0.03496139	Phosphatidylserine lipase ABHD16A
181	Q06187	BTK	−0.32	13.21	0.00292152	0.03496139	Tyrosine-protein kinase BTK
182	Q86V81	ALYREF	0.33	13.65	0.00301063	0.03582983	THO complex subunit 4
183	Q96HC4	PDLIM5	−0.24	13.11	0.00311511	0.03674076	PDZ and LIM domain protein 5
184	P50502	ST13	−0.39	14.49	0.00313602	0.03674076	Hsc70-interacting protein
185	O75874	IDH1	−0.18	13.26	0.00313806	0.03674076	Isocitrate dehydrogenase [NADP] cytoplasmic
186	Q9UIQ6	LNPEP	−0.26	13.04	0.00316421	0.03684776	Leucyl-cystinyl aminopeptidase
187	P05204	HMGN2	0.53	13.65	0.0031931	0.03698535	Non-histone chromosomal protein HMG-17
188	Q9UBE0	SAE1	1.77	13.38	0.00325404	0.03749071	SUMO-activating enzyme subunit 1
189	Q27J81	INF2	−0.25	13.29	0.00329449	0.03764605	Inverted formin-2
190	P17568	NDUFB7	0.9	13.5	0.00330229	0.03764605	NADH dehydrogenase [ubiquinone] 1 beta subcomplex subunit 7
191	P42226	STAT6	0.47	12.82	0.0033913	0.03818853	Signal transducer and activator of transcription 6
192	Q96A72	MAGOHB	0.51	13.65	0.00339134	0.03818853	Protein mago nashi homolog 2
193	Q969T9	WBP2	−0.44	15.05	0.00340276	0.03818853	WW domain-binding protein 2
194	P25098	GRK2	−0.24	13.1	0.00345488	0.03857356	Beta-adrenergic receptor kinase 1
195	Q9UHA4	LAMTOR3	0.38	12.83	0.00352314	0.03898614	Ragulator complex protein LAMTOR3
196	O75558	STX11	−0.26	13.12	0.00352783	0.03898614	Syntaxin-11
197	Q9NS28	RGS18	0.66	13.35	0.00357069	0.03925947	Regulator of G-protein signaling 18
198	P07737	PFN1	−0.38	16.53	0.0036122	0.03951523	Profilin-1
199	P13807	GYS1	0.9	13.13	0.00363219	0.03953429	Glycogen [starch] synthase, muscle
200	P28838	LAP3	−0.28	13.87	0.00382286	0.04140152	Cytosol aminopeptidase
201	Q8NBS9	TXNDC5	−0.27	13.77	0.00390772	0.0421101	Thioredoxin domain-containing protein 5
202	P04114	APOB	−0.37	13.83	0.00396257	0.04248971	Apolipoprotein B-100
203	Q92597	NDRG1	0.69	13.96	0.00401989	0.04289198	Protein NDRG1
204	Q10472	GALNT1	1.1	13.6	0.00406443	0.04315433	Polypeptide N-acetylgalactosaminyltransferase 1
205	P16930	FAH	−0.36	13.13	0.00408432	0.04315433	Fumarylacetoacetase
206	Q9Y2T2	AP3M1	−0.35	13.23	0.00410891	0.04320338	AP-3 complex subunit mu-1
207	Q01813	PFKP	−0.2	13.41	0.00416037	0.04353312	ATP-dependent 6-phosphofructokinase, platelet type
208	Q00577	PURA	0.69	13.2	0.00419225	0.04363857	Transcriptional activator protein Pur-alpha
209	O15143	ARPC1B	−0.22	14.67	0.00421074	0.04363857	Actin-related protein 2/3 complex subunit 1B
210	Q7KZF4	SND1	−0.16	13.84	0.00434483	0.04458141	Staphylococcal nuclease domain-containing protein 1
211	Q7L576	CYFIP1	−0.23	13.53	0.00436386	0.04458141	Cytoplasmic *FMR1*-interacting protein 1
212	Q3ZCW2	LGALSL	−0.29	13.07	0.00436939	0.04458141	Galectin-related protein
213	Q8N8A2	ANKRD44	0.71	13.04	0.00438404	0.04458141	Serine/threonine-protein phosphatase 6 regulatory ankyrin repeat subunit B
214	Q5RKV6	EXOSC6	0.88	13.39	0.00464216	0.04698563	Exosome complex component MTR3
215	P61952	GNG11	1.69	14.01	0.00471723	0.04752336	Guanine nucleotide-binding protein G(I)/G(S)/G(O) subunit gamma-11
216	P16157	ANK1	−0.43	13.09	0.0048462	0.04859658	Ankyrin-1
217	Q9NZ01	TECR	0.94	13.35	0.00488912	0.04880109	Very-long-chain enoyl-CoA reductase
218	Q14644	RASA3	−0.18	13.25	0.00493951	0.04886889	Ras GTPase-activating protein 3
219	Q13813	SPTAN1	−0.2	13.12	0.00495471	0.04886889	Spectrin alpha chain, non-erythrocytic 1
220	Q9Y262	EIF3L	0.25	13.39	0.0049636	0.04886889	Eukaryotic translation initiation factor 3 subunit L
221	P62701	RPS4X	0.19	13.69	0.00499505	0.04895604	40S ribosomal protein S4, X isoform
222	Q8N4P3	HDDC3	0.35	12.82	0.005064	0.04901974	Guanosine-3′,5′-bis(diphosphate) 3′-pyrophosphohydrolase MESH1
223	O75165	DNAJC13	−0.25	12.88	0.00506932	0.04901974	DnaJ homolog subfamily C member 13
224	Q9P2X0	DPM3	0.8	13.05	0.00506945	0.04901974	Dolichol-phosphate mannosyltransferase subunit 3
225	Q13057	COASY	−0.25	12.88	0.00516473	0.04971913	Bifunctional coenzyme A synthase
226	Q9Y3L3	SH3BP1	−0.32	13.55	0.00523176	0.04993232	SH3 domain-binding protein 1
227	O75083	WDR1	−0.17	14.37	0.00523298	0.04993232	WD repeat-containing protein 1

**Table 4 ijms-24-13477-t004:** Differentially expressed proteins between converter and non-converter at Visit 2.

Sr #	Protein Accessions	Genes	logFC	AveExpr	*p*-Value	adj.P.Val	Protein Descriptions
1	P42025	ACTR1B	0.79	12.99	9.15 × 10^−9^	1.98 × 10^−5^	Beta-centractin
2	P02656	APOC3	2.03	14.74	1.86 × 10^−7^	0.000202	Apolipoprotein C-III
3	P12829	MYL4	1.63	13.41	3.63 × 10^−7^	0.000262	Myosin light chain 4
4	Q08380	LGALS3BP	1.16	13.2	5.15 × 10^−7^	0.000279	Galectin-3-binding protein
5	Q8N386	LRRC25	1.19	13.33	6.85 × 10^−7^	0.000297	Leucine-rich repeat-containing protein 25
6	O95168	NDUFB4	2.2	14.47	9.16 × 10^−7^	0.000331	NADH dehydrogenase [ubiquinone] 1 beta subcomplex subunit 4
7	P08697	SERPINF2	2.05	15.88	1.69 × 10^−6^	0.000408	Alpha-2-antiplasmin
8	Q02108	GUCY1A1	0.59	12.9	1.82 × 10^−6^	0.000408	Guanylate cyclase soluble subunit alpha-1
9	Q15370	ELOB	0.99	13.11	1.85 × 10^−6^	0.000408	Elongin-B
10	O00483	NDUFA4	2.45	14.82	1.88 × 10^−6^	0.000408	Cytochrome c oxidase subunit NDUFA4
11	Q92597	NDRG1	1.15	13.96	3.48 × 10^−6^	0.000685	Protein NDRG1
12	P51809	VAMP7	0.56	12.94	4.36 × 10^−6^	0.000772	Vesicle-associated membrane protein 7
13	P04424	ASL	0.6	13.02	4.63 × 10^−6^	0.000772	Argininosuccinate lyase
14	P04217	A1BG	2.36	16.03	5.69 × 10^−6^	0.000814	Alpha-1B-glycoprotein
15	P30048	PRDX3	0.82	14.71	5.84 × 10^−6^	0.000814	Thioredoxin-dependent peroxide reductase, mitochondrial
16	P04350	TUBB4A	0.68	12.94	6.23 × 10^−6^	0.000814	Tubulin beta-4A chain
17	P51570	GALK1	0.97	13.2	6.39 × 10^−6^	0.000814	Galactokinase
18	A6NHR9	SMCHD1	−0.32	13.09	7.35 × 10^−6^	0.000885	Structural maintenance of chromosomes flexible hinge domain-containing protein 1
19	A8MWD9; P62308	SNRPGP15; SNRPG	0.96	14.17	9.53 × 10^−6^	0.001087	Putative small nuclear ribonucleoprotein G-like protein 15; Small nuclear ribonucleoprotein G
20	P18669	PGAM1	0.79	14.77	1.05 × 10^−5^	0.001132	Phosphoglycerate mutase 1
21	P05543	SERPINA7	1.35	13.38	1.42 × 10^−5^	0.001465	Thyroxine-binding globulin
22	P49065	ALB	2.01	15.74	1.79 × 10^−5^	0.001758	Serum albumin
23	Q02750	MAP2K1	2.64	15.39	1.89 × 10^−5^	0.001758	Dual specificity mitogen-activated protein kinase kinase 1
24	Q00577	PURA	1.05	13.2	1.95 × 10^−5^	0.001758	Transcriptional activator protein Pur-alpha
25	Q9H8H3	METTL7A	1.36	14.82	2.27 × 10^−5^	0.001967	Methyltransferase-like protein 7A
26	Q8IZ07	ANKRD13A	0.92	12.9	2.58 × 10^−5^	0.002148	Ankyrin repeat domain-containing protein 13A
27	Q92769	HDAC2	0.5	12.95	2.74 × 10^−5^	0.002199	Histone deacetylase 2
28	Q92530	PSMF1	0.57	12.9	3.55 × 10^−5^	0.002683	Proteasome inhibitor PI31 subunit
29	Q86X76	NIT1	1.32	13.35	3.59 × 10^−5^	0.002683	Deaminated glutathione amidase
30	P55795	HNRNPH2	0.72	14.1	3.79 × 10^−5^	0.002735	Heterogeneous nuclear ribonucleoprotein H2
31	C4AMC7; Q6VEQ5	WASH3P; WASH2P	0.88	13.21	3.94 × 10^−5^	0.002751	Putative WAS protein family homolog 3; WAS protein family homolog 2
32	O95182	NDUFA7	1.37	13.78	4.53 × 10^−5^	0.003066	NADH dehydrogenase [ubiquinone] 1 alpha subcomplex subunit 7
33	P10412	HIST1H1E	−0.54	17.51	4.71 × 10^−5^	0.003089	Histone H1.4
34	P05387	RPLP2	0.81	14.33	4.88 × 10^−5^	0.003108	60S acidic ribosomal protein P2
35	Q06587	RING1	0.76	12.94	6.84 × 10^−5^	0.004089	E3 ubiquitin-protein ligase RING1
36	Q8NFV4	ABHD11	0.46	12.85	6.90 × 10^−5^	0.004089	Protein ABHD11
37	P16401	HIST1H1B	−0.69	16.82	6.98 × 10^−5^	0.004089	Histone H1.5
38	P17568	NDUFB7	1.23	13.5	7.45 × 10^−5^	0.004168	NADH dehydrogenase [ubiquinone] 1 beta subcomplex subunit 7
39	Q8NHV1	GIMAP7	0.78	13	7.50 × 10^−5^	0.004168	GTPase IMAP family member 7
40	P15153	RAC2	0.86	14.68	8.65 × 10^−5^	0.004687	Ras-related C3 botulinum toxin substrate 2
41	P31483	TIA1	0.63	12.94	0.000107	0.005643	Nucleolysin TIA-1 isoform p40
42	P27986	PIK3R1	0.33	12.91	0.000114	0.005884	Phosphatidylinositol 3-kinase regulatory subunit alpha
43	Q13363	CTBP1	0.43	13.34	0.000138	0.006932	C-terminal-binding protein 1
44	Q6DD87	ZNF787	0.52	12.84	0.000149	0.007156	Zinc finger protein 787
45	O95379	TNFAIP8	0.6	13.02	0.000156	0.007156	Tumor necrosis factor alpha-induced protein 8
46	P56279	TCL1A	1.27	13.06	0.000158	0.007156	T-cell leukemia/lymphoma protein 1A
47	Q9H9G7; Q9HCK5; Q9UL18	AGO3; AGO4; AGO1	1.58	13.52	0.000159	0.007156	Protein argonaute-3; Protein argonaute-4; Protein argonaute-1
48	O00505	KPNA3	1.02	13.4	0.000159	0.007156	Importin subunit alpha-4
49	P30043	BLVRB	0.67	13.49	0.000168	0.00736	Flavin reductase (NADPH)
50	Q9P2R7	SUCLA2	−0.47	14.11	0.00017	0.00736	Succinate-CoA ligase [ADP-forming] subunit beta, mitochondrial
51	Q9NUQ9	FAM49B	0.36	13.56	0.000175	0.00736	Protein FAM49B
52	Q7Z4Q2	HEATR3	0.41	12.81	0.000179	0.00736	HEAT repeat-containing protein 3
53	Q9Y6W5	WASF2	0.66	14.35	0.00018	0.00736	Wiskott-Aldrich syndrome protein family member 2
54	Q93050	ATP6V0A1	0.55	12.82	0.000184	0.007376	V-type proton ATPase 116 kDa subunit a isoform 1
55	P43304	GPD2	−0.29	13.5	0.000187	0.007376	Glycerol-3-phosphate dehydrogenase, mitochondrial
56	Q16630	CPSF6	0.4	13.58	0.000201	0.007785	Cleavage and polyadenylation specificity factor subunit 6
57	Q96NY7; Q9NZA1	CLIC6; CLIC5	1.92	13.95	0.000213	0.008057	Chloride intracellular channel protein 6; Chloride intracellular channel protein 5
58	P46379	BAG6	0.94	13.18	0.000218	0.008057	Large proline-rich protein BAG6
59	B2RUZ4	SMIM1	−0.54	13.36	0.000219	0.008057	Small integral membrane protein 1
60	Q06323	PSME1	0.5	14.2	0.000248	0.008953	Proteasome activator complex subunit 1
61	P02760	AMBP	1.11	16.26	0.000257	0.009141	Protein AMBP
62	Q15907	RAB11B	−0.39	14.99	0.000266	0.009285	Ras-related protein Rab-11B
63	Q00059	TFAM	−0.35	13.4	0.000298	0.010133	Transcription factor A, mitochondrial
64	Q15287	RNPS1	1.83	14.51	0.000299	0.010133	RNA-binding protein with serine-rich domain 1
65	Q13813	SPTAN1	−0.26	13.12	0.000325	0.01077	Spectrin alpha chain, non-erythrocytic 1
66	P14678; P63162	SNRPB; SNRPN	0.79	14.22	0.000328	0.01077	Small nuclear ribonucleoprotein-associated proteins B and B’; Small nuclear ribonucleoprotein-associated protein N
67	P02452	COL1A1	1.29	14.38	0.000354	0.011289	Collagen alpha-1(I) chain
68	P05452	CLEC3B	1.23	14.08	0.000354	0.011289	Tetranectin
69	Q8NG11	TSPAN14	−0.44	13.54	0.000376	0.011791	Tetraspanin-14
70	P26641	EEF1G	−0.38	14.6	0.000399	0.012224	Elongation factor 1-gamma
71	O75746	SLC25A12	−0.3	13.14	0.000401	0.012224	Calcium-binding mitochondrial carrier protein Aralar1
72	Q9Y333	LSM2	−0.37	12.85	0.000409	0.012267	U6 snRNA-associated Sm-like protein LSm2
73	P42285	MTREX	0.5	12.85	0.000413	0.012267	Exosome RNA helicase MTR4
74	P62318	SNRPD3	1.02	15.6	0.00047	0.013767	Small nuclear ribonucleoprotein Sm D3
75	Q9BUJ2	HNRNPUL1	0.42	13.81	0.000492	0.014049	Heterogeneous nuclear ribonucleoprotein U-like protein 1
76	Q8WXF7	ATL1	0.4	12.87	0.000499	0.014049	Atlastin-1
77	P20933	AGA	1.16	14.15	0.000499	0.014049	N(4)-(beta-N-acetylglucosaminyl)-L-asparaginase
78	P28838	LAP3	−0.34	13.87	0.000512	0.014231	Cytosol aminopeptidase
79	P32942	ICAM3	−0.57	14.85	0.00052	0.014251	Intercellular adhesion molecule 3
80	Q96BM9	ARL8A	−0.42	14.46	0.00053	0.014361	ADP-ribosylation factor-like protein 8A
81	Q9H098	FAM107B	1.88	13.65	0.00058	0.015505	Protein FAM107B
82	Q8N699	MYCT1	1.13	13.32	0.000589	0.015554	Myc target protein 1
83	O75116	ROCK2	−0.24	13.28	0.0006	0.01557	Rho-associated protein kinase 2
84	Q9UIA9	XPO7	2.07	14.44	0.000607	0.01557	Exportin-7
85	P34910	EVI2B	0.94	13.6	0.000611	0.01557	Protein EVI2B
86	O14980	XPO1	−0.21	12.95	0.000627	0.015784	Exportin-1
87	P23743	DGKA	−0.31	13.18	0.000638	0.015889	Diacylglycerol kinase alpha
88	P14543	NID1	−0.35	13.68	0.000682	0.016787	Nidogen-1
89	Q9UIQ6	LNPEP	−0.3	13.04	0.00072	0.017408	Leucyl-cystinyl aminopeptidase
90	Q32P28	P3H1	0.5	12.87	0.00073	0.017408	Prolyl 3-hydroxylase 1
91	P06730	EIF4E	0.77	13.27	0.000738	0.017408	Eukaryotic translation initiation factor 4E
92	Q15008	PSMD6	−0.22	12.96	0.000747	0.017408	26S proteasome non-ATPase regulatory subunit 6
93	P10599	TXN	0.45	13.66	0.000747	0.017408	Thioredoxin
94	O95299	NDUFA10	−0.23	12.9	0.000778	0.017795	NADH dehydrogenase [ubiquinone] 1 alpha subcomplex subunit 10, mitochondrial
95	Q52LJ0	FAM98B	0.7	13.59	0.00078	0.017795	Protein FAM98B
96	O95819	MAP4K4	0.55	14.63	0.000791	0.01784	Mitogen-activated protein kinase kinase kinase kinase 4
97	Q8IV53	DENND1C	0.66	13.19	0.000835	0.018443	DENN domain-containing protein 1C
98	P51884	LUM	2.03	15.33	0.00085	0.018443	Lumican
99	P09917	ALOX5	−0.19	12.9	0.000851	0.018443	Arachidonate 5-lipoxygenase
100	P55957	BID	0.98	13.87	0.000851	0.018443	BH3-interacting domain death agonist
101	Q9BPX5	ARPC5L	0.7	13.69	0.000884	0.018961	Actin-related protein 2/3 complex subunit 5-like protein
102	Q9NS28	RGS18	0.75	13.35	0.000907	0.019263	Regulator of G-protein signaling 18
103	O00193	SMAP	0.38	12.87	0.000947	0.019304	Small acidic protein
104	O95544	NADK	0.84	13.34	0.000948	0.019304	NAD kinase
105	P11908	PRPS2	1.54	13.64	0.000948	0.019304	Ribose-phosphate pyrophosphokinase 2
106	Q13057	COASY	−0.3	12.88	0.000955	0.019304	Bifunctional coenzyme A synthase
107	Q9BY77	POLDIP3	0.64	12.95	0.000957	0.019304	Polymerase delta-interacting protein 3
108	Q86YP4	GATAD2A	2.03	14.08	0.000963	0.019304	Transcriptional repressor p66-alpha
109	P17900	GM2A	0.95	13.05	0.00098	0.019473	Ganglioside GM2 activator
110	Q8NI27	THOC2	−0.42	12.97	0.00102	0.019964	THO complex subunit 2
111	P60660	MYL6	−0.35	15.44	0.001039	0.019964	Myosin light polypeptide 6
112	Q13045	FLII	−0.2	13.48	0.001044	0.019964	Protein flightless-1 homolog
113	P14174	MIF	1.84	15.48	0.001049	0.019964	Macrophage migration inhibitory factor
114	Q86Y39	NDUFA11	0.48	13.48	0.001051	0.019964	NADH dehydrogenase [ubiquinone] 1 alpha subcomplex subunit 11
115	P12109	COL6A1	1.66	14.75	0.001074	0.020052	Collagen alpha-1(VI) chain
116	Q08722	CD47	−1.21	15.13	0.001077	0.020052	Leukocyte surface antigen CD47
117	Q9NVZ3	NECAP2	0.6	13.39	0.001083	0.020052	Adaptin ear-binding coat-associated protein 2
118	Q9BRA2	TXNDC17	0.65	13.95	0.001121	0.020576	Thioredoxin domain-containing protein 17
119	Q16401	PSMD5	−0.55	12.83	0.001208	0.021994	26S proteasome non-ATPase regulatory subunit 5
120	P14625	HSP90B1	−0.22	13.99	0.001298	0.023421	Endoplasmin
121	Q8NCG7	DAGLB	0.48	12.87	0.001353	0.024214	Sn1-specific diacylglycerol lipase beta
122	Q15796	SMAD2	0.84	13.29	0.00138	0.024501	Mothers against decapentaplegic homolog 2
123	P01911	HLA-DRB1	0.66	14.51	0.001503	0.02647	HLA class II histocompatibility antigen, DRB1-15 beta chain
124	P49773	HINT1	0.34	13.97	0.001603	0.027994	Histidine triad nucleotide-binding protein 1
125	P84243	H3-3A	0.71	12.9	0.001653	0.028452	Histone H3.3
126	Q8IYM9	TRIM22	−0.31	12.82	0.001655	0.028452	E3 ubiquitin-protein ligase TRIM22
127	Q15056	EIF4H	0.6	13.81	0.001734	0.029567	Eukaryotic translation initiation factor 4H
128	P04259	KRT6B	1.79	13.9	0.00183	0.030966	Keratin, type II cytoskeletal 6B
129	Q13576	IQGAP2	−0.22	13.26	0.001865	0.031298	Ras GTPase-activating-like protein IQGAP2
130	Q93009	USP7	−0.27	13.32	0.001892	0.031298	Ubiquitin carboxyl-terminal hydrolase 7
131	P09543	CNP	−0.34	12.78	0.001893	0.031298	2′,3′-cyclic-nucleotide 3′-phosphodiesterase
132	P54709	ATP1B3	0.36	13.19	0.001927	0.031614	Sodium/potassium-transporting ATPase subunit beta-3
133	Q8N4P3	HDDC3	0.38	12.82	0.001961	0.031932	Guanosine-3′,5′-bis(diphosphate) 3′-pyrophosphohydrolase MESH1
134	P68402	PAFAH1B2	0.41	12.91	0.002019	0.0324	Platelet-activating factor acetylhydrolase IB subunit beta
135	P22694	PRKACB	0.62	13.1	0.002019	0.0324	cAMP-dependent protein kinase catalytic subunit beta
136	P39687	ANP32A	0.36	14.59	0.002071	0.032871	Acidic leucine-rich nuclear phosphoprotein 32 family member A
137	Q9H3G5	CPVL	0.39	13.61	0.002079	0.032871	Probable serine carboxypeptidase CPVL
138	P62304	SNRPE	0.43	12.95	0.002122	0.033305	Small nuclear ribonucleoprotein E
139	P02749	APOH	0.78	14.82	0.002154	0.033305	Beta-2-glycoprotein 1
140	Q8N5M9	JAGN1	0.44	13.09	0.002171	0.033305	Protein jagunal homolog 1
141	Q6IAA8	LAMTOR1	−0.3	12.72	0.00218	0.033305	Ragulator complex protein LAMTOR1
142	Q9Y3B7	MRPL11	0.74	13.36	0.002183	0.033305	39S ribosomal protein L11, mitochondrial
143	Q96JB5	CDK5RAP3	1.37	13.69	0.002288	0.034652	CDK5 regulatory subunit-associated protein 3
144	P18583	SON	1.1	13.47	0.00231	0.034748	Protein SON
145	Q9Y2T2	AP3M1	−0.36	13.23	0.002378	0.035172	AP-3 complex subunit mu-1
146	P49327	FASN	−0.22	13.12	0.00238	0.035172	Fatty acid synthase
147	O14735	CDIPT	1.07	13.53	0.002395	0.035172	CDP-diacylglycerol-inositol 3-phosphatidyltransferase
148	P50851	LRBA	−0.23	13.29	0.002403	0.035172	Lipopolysaccharide-responsive and beige-like anchor protein
149	Q86VM9	ZC3H18	−0.36	12.91	0.002455	0.035476	Zinc finger CCCH domain-containing protein 18
150	Q9NZK5	ADA2	−0.19	13.15	0.002466	0.035476	Adenosine deaminase 2
151	P01042	KNG1	0.74	13.86	0.002473	0.035476	Kininogen-1
152	P46926	GNPDA1	−0.31	12.92	0.002496	0.035565	Glucosamine-6-phosphate isomerase 1
153	Q9BUQ8	DDX23	−0.39	12.96	0.00253	0.03569	Probable ATP-dependent RNA helicase DDX23
154	Q92506	HSD17B8	−0.56	13.61	0.002538	0.03569	Estradiol 17-beta-dehydrogenase 8
155	Q9P035	HACD3	−0.27	13.02	0.002585	0.036072	Very-long-chain (3R)-3-hydroxyacyl-CoA dehydratase 3
156	O43670	ZNF207	0.71	13.66	0.002598	0.036072	BUB3-interacting and GLEBS motif-containing protein ZNF207
157	P16109	SELP	−0.71	14.99	0.002692	0.037136	P-selectin
158	P07996	THBS1	−0.44	15.63	0.002759	0.037818	Thrombospondin-1
159	P51148	RAB5C	0.61	13.94	0.002839	0.038672	Ras-related protein Rab-5C
160	P30046	DDT	0.32	13.57	0.002967	0.040172	D-dopachrome decarboxylase
161	Q9BTT0	ANP32E	0.44	13.67	0.002997	0.040324	Acidic leucine-rich nuclear phosphoprotein 32 family member E
162	P01909	HLA-DQA1	0.3	12.9	0.003031	0.040522	HLA class II histocompatibility antigen, DQ alpha 1 chain
163	Q9H4I9	SMDT1	0.68	13.48	0.003092	0.041051	Essential MCU regulator, mitochondrial
164	Q9Y5Z4	HEBP2	0.21	13.05	0.003108	0.041051	Heme-binding protein 2
165	P42126	ECI1	0.6	13.03	0.003131	0.041095	Enoyl-CoA delta isomerase 1, mitochondrial
166	Q9C0C9	UBE2O	−0.24	13.08	0.003192	0.041121	(E3-independent) E2 ubiquitin-conjugating enzyme
167	P62857	RPS28	0.67	15.39	0.003194	0.041121	40S ribosomal protein S28
168	Q02338	BDH1	1.52	14.41	0.003195	0.041121	D-beta-hydroxybutyrate dehydrogenase, mitochondrial
169	Q86WV1	SKAP1	1.14	13.51	0.00321	0.041121	Src kinase-associated phosphoprotein 1
170	P28072	PSMB6	0.34	13.13	0.003227	0.041121	Proteasome subunit beta type-6
171	O75947	ATP5PD	−0.31	14.52	0.003294	0.04173	ATP synthase subunit d, mitochondrial
172	P01344	IGF2	1.26	14.88	0.003353	0.042227	Insulin-like growth factor II
173	Q96CX2	KCTD12	0.61	14.27	0.003415	0.042752	BTB/POZ domain-containing protein KCTD12
174	P16150	SPN	−0.66	15.38	0.003444	0.042806	Leukosialin
175	P07477	PRSS1	2.06	17.29	0.003458	0.042806	Trypsin-1
176	Q8IV08	PLD3	0.43	12.92	0.00351	0.043046	Phospholipase D3
177	P62277	RPS13	0.34	14.07	0.003518	0.043046	40S ribosomal protein S13
178	P62195	PSMC5	−0.28	12.86	0.003549	0.043146	26S proteasome regulatory subunit 8
179	O95866	MPIG6B	1.07	16.4	0.003566	0.043146	Megakaryocyte and platelet inhibitory receptor G6b
180	Q13761	RUNX3	0.41	12.98	0.003604	0.043363	Runt-related transcription factor 3
181	P08708	RPS17	0.3	13.25	0.003664	0.043527	40S ribosomal protein S17
182	P62330	ARF6	0.87	13.46	0.003688	0.043527	ADP-ribosylation factor 6
183	P25789	PSMA4	0.42	13.69	0.003701	0.043527	Proteasome subunit alpha type-4
184	P12236	SLC25A6	−0.29	15.3	0.003703	0.043527	ADP/ATP translocase 3
185	Q9NQG5	RPRD1B	−0.37	13.01	0.003718	0.043527	Regulation of nuclear pre-mRNA domain-containing protein 1B
186	P20340	RAB6A	−0.35	14.33	0.003816	0.044395	Ras-related protein Rab-6A
187	P17676	CEBPB	1.26	13.57	0.003833	0.044395	CCAAT/enhancer-binding protein beta
188	O60831	PRAF2	0.68	13.17	0.003879	0.044515	PRA1 family protein 2
189	P62140	PPP1CB	0.91	15.63	0.003884	0.044515	Serine/threonine-protein phosphatase PP1-beta catalytic subunit
190	Q86UT6	NLRX1	−0.33	12.84	0.003926	0.044761	NLR family member X1
191	Q9Y3B2	EXOSC1	0.41	12.98	0.003977	0.045099	Exosome complex component CSL4
192	O76074	PDE5A	−0.27	13.36	0.004052	0.045708	cGMP-specific 3′,5′-cyclic phosphodiesterase
193	Q9UII2	ATP5IF1	−0.39	13.87	0.00417	0.046799	ATPase inhibitor, mitochondrial
194	Q99961	SH3GL1	−0.29	13.41	0.004231	0.047238	Endophilin-A2
195	O00487	PSMD14	0.88	13.34	0.004419	0.049082	26S proteasome non-ATPase regulatory subunit 14
196	Q96K37	SLC35E1	−0.35	12.86	0.004464	0.049327	Solute carrier family 35 member E1

**Table 5 ijms-24-13477-t005:** Differentially expressed proteins between converter and non-converter at Visit 1 and Visit 2.

Sr #	Protein Accessions	Genes	logFC.V1	AveExpr.V1	logFC.V2	AveExpr.V2	Protein Descriptions
1	Q9Y3B2	EXOSC1	0.848601	12.98093	0.406504	12.9809302	Exosome complex component CSL4
2	P55957	BID	1.594553	13.86545	0.98241	13.8654541	BH3-interacting domain death agonist
3	Q15370	ELOB	1.001451	13.10707	0.993034	13.1070734	Elongin-B
4	P17676	CEBPB	2.159903	13.57377	1.255927	13.5737735	CCAAT/enhancer-binding protein beta
5	Q8NHV1	GIMAP7	0.904981	12.99537	0.77902	12.9953678	GTPase IMAP family member 7
6	Q92769	HDAC2	0.523821	12.94668	0.498368	12.9466806	Histone deacetylase 2
7	Q9BY77	POLDIP3	0.874112	12.94939	0.640527	12.949388	Polymerase delta-interacting protein 3
8	P01909	HLA-DQA1	0.455705	12.89788	0.298861	12.8978826	HLA class II histocompatibility antigen, DQ alpha 1 chain
9	P02656	APOC3	1.57864	14.73972	2.034025	14.7397213	Apolipoprotein C-III
10	P30048	PRDX3	0.751728	14.71094	0.821021	14.7109441	Thioredoxin-dependent peroxide reductase, mitochondrial
11	P62857	RPS28	0.988089	15.38677	0.670676	15.3867658	40S ribosomal protein S28
12	Q02750	MAP2K1	2.535592	15.39421	2.643298	15.394209	Dual specificity mitogen-activated protein kinase kinase 1
13	Q8IV08	PLD3	0.62771	12.92078	0.43252	12.9207843	Phospholipase D3
14	P51148	RAB5C	0.857086	13.93679	0.60824	13.9367931	Ras-related protein Rab-5C
15	O14980	XPO1	−0.255908	12.95095	−0.211783	12.9509484	Exportin-1
16	Q02108	GUCY1A1	0.485112	12.90377	0.58852	12.9037683	Guanylate cyclase soluble subunit alpha-1
17	Q7Z4Q2	HEATR3	0.44294	12.80701	0.408966	12.8070147	HEAT repeat-containing protein 3
18	P42025	ACTR1B	0.515136	12.99252	0.79344	12.9925248	Beta-centractin
19	Q86WV1	SKAP1	1.573969	13.50938	1.136388	13.5093818	Src kinase-associated phosphoprotein 1
20	Q8N699	MYCT1	1.3055	13.31634	1.126395	13.3163412	Myc target protein 1
21	O00483	NDUFA4	1.946313	14.82427	2.44735	14.8242724	Cytochrome c oxidase subunit NDUFA4
22	Q86YP4	GATAD2A	2.446766	14.07985	2.030024	14.0798512	Transcriptional repressor p66-alpha
23	O75116	ROCK2	−0.269602	13.27577	−0.235161	13.275773	Rho-associated protein kinase 2
24	P04350	TUBB4A	0.556699	12.93662	0.676363	12.936615	Tubulin beta-4A chain
25	Q13363	CTBP1	0.418212	13.33699	0.43115	13.3369899	C-terminal-binding protein 1
26	P42285	MTREX	0.523091	12.85032	0.497444	12.8503171	Exosome RNA helicase MTR4
27	O00193	SMAP	0.432867	12.86767	0.384342	12.8676676	Small acidic protein
28	P46379	BAG6	0.910913	13.18182	0.940548	13.1818226	Large proline-rich protein BAG6
29	O00487	PSMD14	1.117681	13.3401	0.879198	13.3401005	26S proteasome non-ATPase regulatory subunit 14
30	P04424	ASL	0.447448	13.02412	0.598387	13.0241187	Argininosuccinate lyase
31	Q96K37	SLC35E1	−0.442733	12.86068	−0.348354	12.860683	Solute carrier family 35 member E1
32	Q9UIA9	XPO7	2.14244	14.44199	2.074225	14.4419886	Exportin-7
33	O95819	MAP4K4	0.57493	14.63426	0.551376	14.6342571	Mitogen-activated protein kinase kinase kinase kinase 4
34	Q8NFV4	ABHD11	0.394209	12.85493	0.461129	12.8549305	Protein ABHD11
35	Q8NCG7	DAGLB	0.515791	12.86969	0.478891	12.8696866	Sn1-specific diacylglycerol lipase beta
36	Q9Y3B7	MRPL11	0.833267	13.35751	0.744418	13.3575081	39S ribosomal protein L11, mitochondrial
37	P12829	MYL4	1.011231	13.40768	1.625336	13.40768	Myosin light chain 4
38	Q99961	SH3GL1	−0.341088	13.4084	−0.286379	13.4084045	Endophilin-A2
39	O60831	PRAF2	0.793311	13.16934	0.67582	13.1693404	PRA1 family protein 2
40	O95182	NDUFA7	1.092824	13.77569	1.368828	13.7756949	NADH dehydrogenase [ubiquinone] 1 alpha subcomplex subunit 7
41	O95168	NDUFB4	1.403307	14.4717	2.198141	14.4716973	NADH dehydrogenase [ubiquinone] 1 beta subcomplex subunit 4
42	Q96JB5	CDK5RAP3	1.487408	13.68659	1.365529	13.6865904	CDK5 regulatory subunit-associated protein 3
43	Q6DD87	ZNF787	0.449662	12.83611	0.524829	12.8361067	Zinc finger protein 787
44	P68402	PAFAH1B2	0.442696	12.91273	0.414401	12.9127301	Platelet-activating factor acetylhydrolase IB subunit beta
45	Q9P035	HACD3	−0.293977	13.0163	−0.2693	13.0163005	Very-long-chain (3R)-3-hydroxyacyl-CoA dehydratase 3
46	Q9Y333	LSM2	−0.334766	12.85472	−0.365537	12.8547206	U6 snRNA-associated Sm-like protein LSm2
47	Q9C0C9	UBE2O	−0.270949	13.07538	−0.243153	13.075381	(E3-independent) E2 ubiquitin-conjugating enzyme
48	Q15056	EIF4H	0.623605	13.81299	0.603652	13.8129855	Eukaryotic translation initiation factor 4H
49	O95299	NDUFA10	−0.218766	12.90348	−0.228773	12.9034786	NADH dehydrogenase [ubiquinone] 1 alpha subcomplex subunit 10, mitochondrial
50	Q9Y5Z4	HEBP2	0.231251	13.05045	0.21175	13.0504482	Heme-binding protein 2
51	O14735	CDIPT	1.111774	13.53497	1.071985	13.5349659	CDP-diacylglycerol-inositol 3-phosphatidyltransferase
52	Q32P28	P3H1	0.458678	12.86789	0.499696	12.8678948	Prolyl 3-hydroxylase 1
53	P08697	SERPINF2	1.255988	15.87848	2.053464	15.8784812	Alpha-2-antiplasmin
54	Q8WXF7	ATL1	0.348227	12.86694	0.39655	12.8669358	Atlastin-1
55	P01911	HLA-DRB1	0.641901	14.50714	0.663274	14.507142	HLA class II histocompatibility antigen, DRB1-15 beta chain
56	P11908	PRPS2	1.425429	13.63723	1.541265	13.6372317	Ribose-phosphate pyrophosphokinase 2
57	Q86UT6	NLRX1	−0.351884	12.83793	−0.329374	12.8379265	NLR family member X1
58	Q9UIQ6	LNPEP	−0.26494	13.0362	−0.302072	13.0361981	Leucyl-cystinyl aminopeptidase
59	P17568	NDUFB7	0.90106	13.50154	1.231176	13.5015444	NADH dehydrogenase [ubiquinone] 1 beta subcomplex subunit 7
60	Q9NS28	RGS18	0.659701	13.35256	0.74681	13.352561	Regulator of G-protein signaling 18
61	P28838	LAP3	−0.27955	13.87283	−0.335483	13.8728298	Cytosol aminopeptidase
62	Q92597	NDRG1	0.686343	13.95879	1.152069	13.9587948	Protein NDRG1
63	Q9Y2T2	AP3M1	−0.347239	13.2346	−0.363207	13.2345966	AP-3 complex subunit mu-1
64	Q00577	PURA	0.686034	13.19965	1.050156	13.1996452	Transcriptional activator protein Pur-alpha
65	Q13813	SPTAN1	−0.198628	13.121	−0.255255	13.1210042	Spectrin alpha chain, non-erythrocytic 1
66	Q8N4P3	HDDC3	0.351613	12.82065	0.384854	12.8206512	Guanosine-3′,5′-bis(diphosphate) 3′-pyrophosphohydrolase MESH1
67	Q13057	COASY	−0.25482	12.88284	−0.299964	12.8828366	Bifunctional coenzyme A synthase

**Table 6 ijms-24-13477-t006:** Differentially expressed proteins between Visit 1 and Visit 2 in converter and non-converter.

Sr #	Group	Differentially Expressed Proteins
1	Converter	COX5A,MT-CO2,VASP,C3,LSM3,EIF5A,GC,BID,ALB,ZNF207,AHSG,RAB32,UQCRH,F2,CMAS,A2M,SH3BGRL,AFP,SERPINF1,SERPINC1,BANF1,CALM1;CALM2;CALM3,GRB2,SAP18,UQCRQ,WASF2,ISOC1,AHNAK,C4A,ADD3,CNN2,SLTM,HIST1H1E,SF3B2,GLIPR2,FN1,LPCAT3,MTPN,COX7A2,SKP1,ABCC4,RPS26,PRPS1,ITIH2,HBA2,NONO,RAB6A,OGDH,EXOSC1,SNRPF,UQCR10,RAB11B,USP14,PAFAH1B3,ITIH3,RAE1,SLC30A7,U2AF2,RBM8A,COX6B1,GP1BA,WARS,GIMAP4,DDT,DNAJC13,MYCT1,ARPC3,SMARCC2,ENO2,HCLS1,APOB,PPP1CA,VAT1,RPL31,FBLN1,BLVRA,COL1A1,CAB39,AK2,OSBPL8,CTSB,CNDP2,TPD52L2,LTA4H,TUFM,ARF3;ARF1,ACTG1,PCYT1A,SUCLA2,SNX2,ST13,LAMTOR4,LMAN2,CLEC1B,SYNGR2,RAB18,NDUFA10,PCMT1,PDXK,COL6A1,SARS,ANXA11,NDUFB6,TRAF3IP3,WAS,RAB3D,ZYX,SLC9A3R1,DAD1,UBXN1,TFAM,SASH3,PGK1,TMPO,G3BP1,ALDOA,HM13,RNH1,BIN2,RPL36A;RPL36AL,PSMC2,ACO2,APOH,CEBPB,RPL9P8,TCP1,HNRNPA3,RBX1,PSIP1,GATD3B;GATD3A,PAK2,HSD17B11,HIST1H2BJ,EEF1A1;EEF1A1P5,SCP2,MRE11,COX5B,CHCHD2,IGFBP2,MYL6,NUDC,RO60,PNKD,RAB6B,SART1,PLPBP,DTD1,SRP9,MAGOHB,GART,INPP5A,BAZ1B,COL1A2,MAT2A,ABRACL,CHMP1B,PRDX1,JPT1,HLA-DQA1
2	NonConverter	OSBPL8

## Data Availability

The datasets generated during and/or analyzed during the current study are available from the corresponding author on reasonable request.

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
