# Peer review of "Blood Proteome Profiling Reveals Biomarkers and Pathway Alterations in Fragile X PM at Risk for Developing FXTAS"

_ijms, 2023, doi:10.3390/ijms241713477_

Round 1

Reviewer 1 Report

In the paper by Piotr Gabryel et al, the authors applied longitudinal blood proteomic profiling of premutation carriers to reveale biomarkers and dysregulated pathways associated with FXTAS development. Mitochondrial bioenergetics pathways were altered, supporting FXTAS pathogenesis. Common proteins were found between CSF proteome of the FXTAS patients and premutation carriers who later developed FXTAS, representing novel findings in this field.

Overall, this paper provides a useful resource in FXTAS research and the logic of this article is relatively straightforward. However, I have a few concerns about the selection of methods and interpretation of the results. Below I provide a series of inquiries and recommendations to the authors that hopefully will help improve the manuscript:

Comment1:

In proteomic analysis, missing values are a common issue that often requires imputation methods for data completion. The authors were not clear if they encountered such a situation during their protein profiling analysis. If they did, it would be helpful to provide a detailed description in the methods section regarding the specific imputation techniques utilized to handle missing values.

Comment2:

Regarding differential proteins and pathways, the authors should provide a clearer clarification on whether they used raw p-values or adjusted p-values (e.g., Benjamini-Hochberg or Bonferroni correction). This information is crucial for the readers to understand the statistical significance and control for multiple testing in their analysis.

Comment3:

In author's manuscript, the threshold of selecting differential expressed proteins was solely based on the adjusted P-value. The other statistic values such as fold change did not include in the further analyse. This omission of fold change values in the subsequent analysis of overlapping differentially expressed proteins may introduce analytical challenges. For example, the overlapping DE proteins in V1 might exhibit opposite expression pattens in V2 (up-expressed in V1 while down-expressed in V2). It would be beneficial to investigate whether all DEPs display a consistent pattern. If not, it would be worth investigating the biological mechanisms behind it.

Comment4:

In Figure 6 and 8, what does edges represent in the network plots?

Comment5:

In line 121-123, the authors stated “We identified potential predictive proteomic biomarkers of early diagnosis and reported the altered protein pathways among the groups suggesting their role in the pathogenies of the disorder”. I am not entirely sure if using the term "predictive proteomic biomarkers" is appropriate here, as there is limited discussion about their potential as predictive factors in the article.

Author Response

REVIEWER 1

In the paper by Piotr Gabryel et al, the authors applied longitudinal blood proteomic profiling of premutation carriers to revealed biomarkers and dysregulated pathways associated with FXTAS development. Mitochondrial bioenergetics pathways were altered, supporting FXTAS pathogenesis. Common proteins were found between CSF proteome of the FXTAS patients and premutation carriers who later developed FXTAS, representing novel findings in this field. Overall, this paper provides a useful resource in FXTAS research, and the logic of this article is relatively straightforward. However, I have a few concerns about the selection of methods and interpretation of the results. Below I provide a series of inquiries and recommendations to the authors that hopefully will help improve the manuscript:

Thank you so much for your comments, we are not sure who are the Piotr Gabryel et al.

Comment 1:

In proteomic analysis, missing values are a common issue that often requires imputation methods for data completion. The authors were not clear if they encountered such a situation during their protein profiling analysis. If they did, it would be helpful to provide a detailed description in the methods section regarding the specific imputation techniques utilized to handle missing values.

We have added the information in the manuscript.

Comment 2:

Regarding differential proteins and pathways, the authors should provide a clearer clarification on whether they used raw p-values or adjusted p-values (e.g., Benjamini-Hochberg or Bonferroni correction). This information is crucial for the readers to understand the statistical significance and control for multiple testing in their analysis.

Benjamini-Hochberg approach was used to correct for multiple tests for the differential expression analysis and the significance cutoff of 0.05 was used to select proteins. Multiple test correction was not carried out for pathway enrichment analyses (as substantial overlap in pathway membership introduces a complex dependence structure among tests, while also reducing the effective number of tests) and the raw p-values were used for results.

Comment 3:

In author's manuscript, the threshold of selecting differential expressed proteins was solely based on the adjusted P-value. The other statistic values such as fold change did not include in the further analyses. This omission of fold change values in the subsequent analysis of overlapping differentially expressed proteins may introduce analytical challenges. For example, the overlapping DE proteins in V1 might exhibit opposite expression pattens in V2 (up-expressed in V1 while down-expressed in V2). It would be beneficial to investigate whether all DEPs display a consistent pattern. If not, it would be worth investigating the biological mechanisms behind it.

We have investigated the V1 and V2 list of significant differentially expressed proteins and their change remain in the same directions between V1 and V2. This can be seen in Table 5.

Comment4:

In Figure 6 and 8, what does edges represent in the network plots?

The edges in the network plots are protein-protein interactions queried from STRING database. This information has been added to the legends of the corresponding plots [Figure 5 and 7].

Comment5:

In line 121-123, the authors stated, “We identified potential predictive proteomic biomarkers of early diagnosis and reported the altered protein pathways among the groups suggesting their role in the pathogenies of the disorder”. I am not entirely sure if using the term "predictive proteomic biomarkers" is appropriate here, as there is limited discussion about their potential as predictive factors in the article.

We have clarified this sentence in the text. We strongly believe that the importance of our study as compared to one published so far resides in the fact that it is a longitudinal study, so we are looking at changes over the time compared to studies that look for difference between target population and controls. We identified changes in expression level of several proteins involved in pathways that are known to be dysregulated in FXTAS (i.e., mitochondrial, lipids) but only in converter group which strongly support their role in this disorder.

Reviewer 2 Report

I have issues with any non-standard abbreviation in the title, as well as PM=premutation carrier.(other papers on this do not use abbreviation [see Hoyos et al., 2017]. Details on the "premutation” in the FMR1 gene have been well explained (though is it really necessary to repat ln 56 in ln 70?), but authors do not elaborate on the fact have FXS but they might have, or may later develop, other fragile X-associated disorders. And this group of disordes has not been discussed at all [and lns 59 to 63 are ideal opportunity.  In addition, people with a premutation can have children. 

Lns 45 - 54 seem to have no relevance, whatsoever. This paragraph should be better linked to FXTAS.

In lns. 57/58 - please, explain - what do you mean by "affect more severely"?

In ln. 64 and on - too little is said on clinical manifestations 

Lns 90 -123 are actually rationale for a hypothesis- this should be less ambiguous - make it succinct and more straig

I, the reader in any case, should easily understand the "resulrs" section  - unfortunately, that is not the case; lnns 126... are pretty confusing in his end. Perhaps y9u should consider UpSet diagram insted of fig 1 [&2]??

Carefully spellcheck!

Author Response

REVIEWER 2

I have issues with any non-standard abbreviation in the title, as well as PM=premutation carrier.(other papers on this do not use abbreviation [see Hoyos et al., 2017]. Details on the "premutation” in the FMR1 gene have been well explained (though is it really necessary to repat ln 56 in ln 70?), but authors do not elaborate on the fact have FXS but they might have, or may later develop, other fragile X-associated disorders. And this group of disordes has not been discussed at all [and lns 59 to 63 are ideal opportunity.  In addition, people with a premutation can have children. 

Lns 45 - 54 seem to have no relevance, whatsoever. This paragraph should be better linked to FXTAS.

This paragraph mentioning the challenges of identifying biomarkers in general in neurodegenerative disorders including the Parkinson and Alzheimer disease and it was used to introduce the late on-set nondegenerative disorder FXTAS.

In lns. 57/58 - please, explain - what do you mean by "affect more severely"?

We have clarified this sentence.

In ln. 64 and on - too little is said on clinical manifestations 

More details are mentioned in the following lanes.

Lns 90 -123 are actually rationale for a hypothesis- this should be less ambiguous - make it succinct and more straig

The paragraphs state the proteomics studies that have been conducted with the obtained results which set the stage for the importance of our study and difference with compared to them (targeted v/s longitudinal).

I, the reader in any case, should easily understand the "resulrs" section  - unfortunately, that is not the case; lnns 126... are pretty confusing in his end. Perhaps y9u should consider UpSet diagram insted of fig 1 [&2]??

We have clarified this sentence.

Comments on the Quality of English Language

Carefully spellcheck!

We have carefully spellchecked throughout the paper.

Round 2

Reviewer 1 Report

 I have nothing more to add and think that the paper is much approved.

Reviewer 2 Report

Much improved.